



# Improvement of hydrological model calibration by selecting multiple parameter ranges

Qiaofeng. Wu[1], Shuguang Liu[1,2], Yi Cai[1,2], Xinjian Li[3] and Yangming Jiang[4]

[1]Department of Hydraulic Engineering, College of Civil Engineering, Tongji University, Shanghai, 200092, China
[2]The Yangtze river water environment key laboratory of the ministry of education, Tongji University, Shanghai, 200092, China
[3]Irrigation Experiment Center of Guangxi Zhuang Autonomous Region, Guilin, 541105, China
[4]Hydrology & Water Resources Bureau of Guangxi Zhuang Autonomous Region, Guilin, 541001, China

Correspondence to: Yi Cai (caiyi@tongji.edu.cn)

**Abstract.** The parameters of hydrological models are usually calibrated to achieve a good performance of the model, owing to the highly non-linear problem of hydrology process modelling. However, parameter calibration efficiency has a direct relation with parameter range. Furthermore, parameter range selection is affected by probability distribution of parameter values, parameter sensitivity and correlation. A newly proposed method is introduced to select and coordinate parameter ranges for improving the calibration of hydrological models with multiple parameters. At first, the probability distribution characteristics of single parameter value was analysed based on 100 samples obtained from independent calibration with initial parameter range and the distribution type (i.e. normal, exponential and uniform distributions) determined for single parameter. Then, the way to select the optimal range for single parameter was demonstrated by comparing different reduced and extended ranges corresponding to the distribution. Next, parameter correlation and sensibility were estimated to coordinate range selection of single parameter and the optimal combination of ranges for all parameters obtained. The results show that the probability of calibrated parameter values of Xinanjiang model takes on the normal or exponential distributions. For normal distribution, selecting the range of high probability density from the initial range is much more efficient for calibration. For exponential distribution, if the initial range can not be extended, selecting the range of high probability density contributes to high objective function. If the initial range can be extended, it is better to make the exponential distribution convert into normal distribution by doubling the range along X-axis direction and subsequently select the range according to normal distribution. Moreover, the coordination of range selection of single parameters makes the calibration of models with multiple parameters more efficient and effective.

**Key words**: hydrological model, calibration, parameter ranges, probability distribution

## 1. Introduction

Hydrological process modelling is an important method for research on water resources management, flood control and disaster mitigation, water conservancy project planning and design, hydrological response to climate change and so on (Zanon et al.,





2010;Papathanasiou et al., 2015). The hydrological model is a type of black-box model in 1932 originally (Sherman, 1932), and conceptual models and distributed models are subsequently put forward in 1960s (Freeze and Harlan, 1969). The three kinds of hydrological models have been significantly improved in recent years and their structures become more and more mature. Theoretically, distributed models have definite physical mechanism of water cycle and all parameters can be measured

in-situ (Abbott et al., 1986;Huang et al., 2014). Conceptual models express hydrological processes in form of some abstract models which come from some physical phenomenon and experience. For example, the interflow and base flow are simplified as the flow from linear reservoirs (Caviedes-Voullième et al., 2012;Lü et al., 2013). As a result, some parameters of conceptual models need calibrating. In general, conceptual models have better performance of modelling the streamflow at catchment outlet than distributed models do, especially for catchments lacking sufficient data (Bao et al., 2010;Cullmann et al., 2011).

Thus, many conceptual models such as HBV model, TOPMODEL, Tank model and Xinanjiang model are of strong vitality (Abebe et al., 2010;Vincendon et al., 2010;Hao et al., 2015). Additionally, the performance of distributed models can be improved after calibration of some parameters. Therefore, all of the hydrological models should be calibrated before engineering applications.

There are two kinds of calibration methods for hydrological models, the trial-error method and auto-calibration method. The

trial-error method depends on plenty of trials for reducing the error of the objective. However, it is difficult to obtain exact optimal solution due to limited enumeration (Boyle et al., 2000). The auto-calibration method is based on stochastic or mathematical methods, having wide application in the non-linear parameter optimization. Compared with the trial-error method, it is more efficient and effective, avoiding the interference of anthropogenic factors (Madsen, 2000;Getirana, 2010). The initial automatic optimization methods, such as the Rosenbrock Method (Rosenbrock, 1960) and the Simplex Method

(Nelder and Mead, 1965), are classical and useful methods, but has its limitation of initial value ranges of parameters. Therefore, it can only be regarded as local optimization algorithms (Gupta and Sorooshian, 1985). Different from classical methods above, the Genetic Algorithm (GA) is of random search strategy that avoids problem of local search, being a global optimization algorithm in a real sense (Wang, 1991, 1997;Sedki et al., 2009;Chandwani et al., 2015). After that, many global optimization algorithms have been proposed inheriting the random search strategy. The Shuffled Complex Evolution (SCE-UA) method

combines many advantages of Genetic Algorithm and Simplex Method, having powerful capability of calibrating the rainfall-runoff model (Duan et al., 1994;Zhang and Shi, 2011). The Particle Swarm Optimization (PSO) based on random solution can directly obtain the identification parameters through the iterative search for the optimal solution (Kennedy, 1997;Zambrano-Bigiarini and Rojas, 2013). Though the auto-calibration method has been intensively employed to calibrate parameters in the field of hydrology, the most advanced algorithm inevitably falls into local solution because of the strong non-linear problem

of the hydrological model and parameter correlation (Chu et al., 2010;Jiang et al., 2010;Jiang et al., 2015).

In general, parameter variables obey some types of probability distribution in the given range after multiple independent repeat calibration by an auto-calibration method (Viola et al., 2009;Jin et al., 2010;Li et al., 2010). Graziani et al. (2008) stated that the shapes of the parameter value probability distributions can be significantly affected by their ranges. Ben et al. (2013) studied the effects of different probability distributions (e.g., Normal distribution and uniform distribution) of parameters





values on parameter sensitivity, and found that the probability distribution can be provide a clue to realize parameter sensitivity. Although Normal and uniform distributions are greatly studied in practice, other types of probability distributions seldom were investigated in previous researches (Kucherenko et al., 2012;Esmaeili et al., 2014).

Most hydrological models contain many parameters of different sensitive characteristics and correlation behaviour. Some researchers believe that the sensitive parameter should be calibrated, but the insensitive parameter can be set as a fixed value by experience (Beck, 1987;Cheng et al., 2006). Inappropriate parameter ranges or fixed values may result in the instability of calibrated results. Furthermore, the range setting of one parameter may influence the calibration of other parameters correlated with it (Song et al., 2015). The model parameter sensitivity analysis has been a growing concern in recent years. Parameters sensitivity varies with catchment characteristic, objective function and parameter ranges (van Griensven et al., 2006). Wang et al. (2013) noted the different parameter ranges lead to changes in parameter sensitivity. Shin et al. (2013) reported that reducing or extending the ranges would affect the parameters sensitivity, making insensitive parameters become sensitive ones or vice versa. Thus, parameter ranges and correlation should be taken into considered when the calibration of multi-parameters models is performed.

Parameter ranges are generally given roughly due to lack of knowledge concerning physical settings of a local catchment (Song et al., 2013;Hao et al., 2015). The more deviation between true ranges and given range, the more instability of calculated results. Appropriate parameter ranges selection is critical for calibrating the model efficiently. However, few literature reported how to select the appropriate parameter range for improving the calibration of hydrological models. Furthermore, the calibration of multiple parameters is more complex due to the parameter sensitivity and correlation. Hence, it is necessary to find a way to coordinate the range settings of all parameters.

Considering the effect of parameter ranges on calibration efficiency of hydrological models, an approach of parameter ranges selection (PRS) is put forward to improve the calibration of hydrological models with multiple parameters. At first, probability distribution characteristics of parameter values were analysed based on the parameter value samples that calibrated by using a GA method. Then the optimal range of single parameter is selected for calibration according to its probability distribution. Finally, parameter correlation and sensitivity were estimated to determine the optimal combination of multiple parameters ranges. The proposed method is expected to be helpful for an effective and efficient calibration of hydrological models with multiple parameters.

## 2. Study area and data collection

The Chaotianhe River catchment is located in the northeast of Guangxi Zhuang Autonomous Region in Southwest China (Fig. 1). The Chaotianhe River is the major tributary of the Lijiang River of well-known karst landscape. The total catchment area is 476.24 km$^2$. The annual precipitation is approximately 1704 mm and 78% precipitation concentrates in flood reason (March to August).The thickness of soil varies in most karst areas tremendously different with space: limestone exposed in some peak-cluster region, 2-10 m thickness clay covered in the depression and valley bottom. In clastic rock mountain areas, the thickness of the soil is usually less than 0.5 m. Thus the soil moisture storage capacity varies significantly with space. Moreover, the



underground rivers are very well developed in the karst area, which makes the flood gather rapidly and recess slowly due to higher underground flow rate.

The daily data concerning precipitation, evaporation and streamflow were collected from national gauging stations for the 5-year period of 1996–2000. Four precipitation stations, one streamflow gauging station and one evaporation station are

selected for the investigation. Areal precipitation was calculated using data from the four precipitation stations by using Thiessen polygon method under GIS environment (Cai et al., 2014). The streamflow gauging station is at the catchment outlet. Some metro-hydrological statistical data of the studied catchment are summarized in Table 1. From 1996–2000, the maximum of streamflow is about 719 $m^3$/d, the minimum 0.53 $m^3$/d and the average is 13.31 $m^3$/d at the outlet. The maximum areal precipitation of the studied catchment varies with year, the value is 235 mm/d of 1996 while 107 mm/d of 2000. The average

streamflow decreases from 14.38 to 11.37 $m^3$/d during the studied period.

## 3. Methodology

### 3.1 Hydrological model selection

The method of parameters ranges selections (PRS) is designed for most of hydrological models. At present, there have been many hydrological models for hydrological processes simulation. Considering the climate characteristics of the study area, the

Xinanjiang model which is suitable for humid regions was chosen to serve as the hydrological model for the investigation. The Xinanjiang model mainly includes three evapotranspiration layers and three runoff components (i.e. surface-, subsurface runoff and groundwater) (Zhao et al., 1980;Zhao, 1992). The surface runoff is routed by the Unit Hydrograph (UH) which is derived from observed streamflow and other runoff components are simplified as linear reservoirs (Ju et al., 2009). With regard to the Xinanjiang model, there are 10 parameters that should be calibrated. The meaning and the common range of the parameters

are given in Table 2 (Lin et al., 2014;Hao et al., 2015). The proposed PRS method is introduced as follows, taking a Xinanjiang model for example.

### 3.2 Probability distribution analysis of calibrated parameter value

### 3.2.1 Sample collection of calibrated parameter value

In theory, the results of calibration by using a stochastic-based auto-calibration method are not completely same but similar in

a reasonable convergence condition, which obey some probability distributions (Jiang et al., 2015). In order to analyse the probability distribution of calibrated parameter values, a stochastic-based auto-calibration is used to calibrate the model, and samples of calibrated parameters values are obtained. As far as the sample size is concerned, 100 samples are enough to estimate the probability distribution of calibrated parameter values, which is deduced from plenty of tests by comparing the similitude of distributions and computing efficiency.

A Genetic Algorithm was selected as the auto-calibration method in the investigation, because GAs are common and widespread used global optimization algorithm based on stochastic and evolutionary optimization technique. Many studies





showed that the evolutionary algorithms could provide equal or better performance than other algorithms (Cooper et al., 1997;Jha et al., 2006;Zhang et al., 2009). The Nash–Sutcliffe efficiency ($E_{NS}$) was chosen as the objective function (Eq. (1)) for GA, which representing agreement between observed and simulated data.

$$E_{NS} = 1 - \frac{\Sigma_{i=1}^{n}(Q_{obs,i} - Q_{sim,i})^2}{\Sigma_{i=1}^{n}(Q_{obs,i} - Q_{mean})^2} \tag{1}$$

where $E_{NS}$ is Nash–Sutcliffe efficiency, $i$ serial number of the step; $n$ total number of observed streamflow data, $Q_{obs,i}$ observed streamflow at step $i$, $Q_{sim,i}$ simulated streamflow at step $i$, and $Q_{mean}$ is mean value of observed streamflow.

### 3.2.2 Determination of probability distribution types

The probability distributions of calibrated parameters values can be determined by using box-plot charts, cumulative frequency curves and frequency histograms. Figure 2 shows the three types of probability distribution based on 100 samples of parameter

values of the Xinanjiang model. The symmetry of the box-plot chart (including box and whiskers) and the length ratio of whisker to the box, the shapes of the cumulative frequency curve and the frequency histogram are important indicators to determine the distribution types. Based on these indicators, three types of probability distribution are listed as follows: (1) Normal distributions, the box and whiskers are symmetrical along Y-axis direction, the length of whiskers longer than the height of the box in box-plot chart (Fig. 2a), the cumulative frequency curve S shaped and the histogram is bell shaped (Fig.

2b); (2) Exponential distributions, the whole chart is distinct asymmetrical in the Y-axis direction which means the average value (small hollow square) deviates from the median value (a centre line in box), the box close to one side where the whisker is extreme shorter than that on the opposite side (Fig. 2a), the cumulative frequency curve is parabola shaped, and the histogram tends to increase or decline gradually (Fig. 2c); (3) Uniform distribution, the box and whiskers are symmetrical along Y-axis direction, the length of whiskers approximates to that of the box (Fig. 2a), the cumulative frequency curve tends a line and the

histogram varies little along x-axis (Fig. 2d).

### 3.3 Parameters ranges selections

### 3.3.1 Single parameter range selection (S-PRS)

In order to improve $E_{NS}$, the initial range of parameter is required adjusting properly. In consideration of the three probability distribution types mentioned above, the different ways to adjust the ranges for parameters are presented in the investigation.

For uniform distribution, it is better to keep initial range due to little influence of the range on calibration results. For normal distribution, the cumulative frequency curve is employed to seek several of reduced ranges with a given cumulative frequency, and the minimum and maximum ranges (namely MINR and MAXR) are obtained as depicted in Fig. 3. The MINR and MAXR represents the ranges of maximum and minimum probability density of parameter values with a given cumulative frequency, respectively. As for exponential distribution, the initial range can be doubled from the boundary of high probability density to

the outside, if the parameter has reasonable meaning in the new range. Thus, the exponential distribution can be converted into





normal distribution and then the optimal range can be selected by using the method for normal distribution. If the initial range can not be extended, the MINR and MAXR are sought out according to the cumulative frequency curve. Through plenty of tests, a cumulative frequency value of 50% was adopted to search the MINR and MAXR, which reduces sampling errors in case of smaller percentage, and increases difference between MINR and MAXR in case of larger percentage. Through

extending or reducing the ranges, the probability distribution of calibrated parameter values can transform and finally convert into approximate uniform distribution.

### 3.3.2 Multiple parameters ranges selections (M-PRS)

In general, there is more or less correlation between parameters for most hydrological models. As far as the Xinanjiang model is concerned, both parameter WM and B refer to the water storage volume – area curve that representing the spatial variability

of soil moisture storage. If the curve is fixed, the larger WM results in the smaller B (Zhao et al., 1980). The change of a parameter range may more or less effect the calibration of other parameters. The correlations among parameters, therefore, should be taken into account, if several parameters ranges require adjusting. If the change of one parameter range has positive influence on calibration of other parameters, the selected ranges for the parameter will contributes to better calibration results. On the contrary, the negative influence may make the contribution of the selected ranges against model calibration. Thus, some

coordination measures should be taken to deal with such contradiction. The index $R_C$ (Eq. (2)) were quantified to analyse the influencing degree of one parameter range change to the calibration of other parameters. The more close value of $R_{C\,Y,X}$ to 1, the greater positive influence of range change of parameter X on calibration of parameter Y. If $R_{C\,Y,X}$ less than 0, it means the negative influence.

$$R_{C\,Y,X} = 1 - \frac{L_{Y,X} - L_{Y,Y}}{L_{Y,Initial} - L_{Y,Y}} \tag{2}$$

Where $R_{C\,Y,X}$ is the influencing degree of range change of parameter X on calibration of parameter Y; $L_{Y,X}$ the range of parameter Y calibrated with selected range of parameter X and initial ranges of other parameters, $L_{Y,Y}$ the range of parameter Y calibrated with selected range of parameter Y and initial ranges of other parameters, and $L_{Y,Initial}$ is the range of parameter Y calibrated with initial ranges of all parameters. The calibrated range of the parameter is calculated except extreme outliers.

If there is negative influence between two parameters, the parameter of high sensitivity is ranked as primary one and its

selected ranges can be kept in the range combination for all parameters, while the initial range is used in place of the selected range to minimize the negative effect for the other parameter of low sensitivity. It is due to the fact that sensitive parameters play more important roles than insensitive parameters do during multi-parameter calibration. In order to assess the sensitivity of parameter range change to $E_{NS}$, index $S_E$ as expressed in Eq. (3) is computed by performing S-PRS method on each parameter. The larger $R_E$, the more concentrated $E_{NS}$ distribution, which means parameter calibration is stable and efficient. Thus, the

parameter of high $S_E$ is given priority to use the selected range when $R_C$ of two parameters is minus.

$$S_E = 1 - \frac{E'_{NS\,Max} - E'_{NS\,Min}}{E_{NS\,Max} - E_{NS\,Min}} \tag{3}$$



where $S_E$ is sensitivity of parameter range change to $E_{NS}$, $E_{NS\,Max}$ and $E_{NS\,Min}$ maximum and minimum $E_{NS}$ calibrated with initial range, and $E'_{NS\,Max}$ and $E'_{NS\,Min}$ are maximum and minimum $E_{NS}$ calibrated with selected range. The statistic analysis of $E_{NS}$ excludes extreme outliers.

Considering there are more than two parameters in most hydrological models, the accumulative influence and the coordination of range selection are investigated in the study. Parameters of positive influence on other parameters can be taken into account, while the selected ranges is substituted for the initial ranges for the parameters of negative influence. The mean value of $R_C$ ($R_{C\,mean}$) is the index to judge the accumulative influence of one parameter range change on the calibration of the other parameters. Thus, for parameters of negative $R_{C\,mean}$, the initial ranges instead of the selected one is adopted for calibration of multiple parameters.

The flow chart of the parameter range selection method is shown in Fig. 4. In stage 1, a set of initial parameter ranges is given for a hydrological model and the probability distribution analysed based on the 100 independent parameters values calibrated by an auto-calibration method. In stage 2, there are three range adjustment methods with response to parameter value probability distribution: for normal distribution, the optimal range for single parameter is obtained by reducing the initial range; for exponential distribution, the initial range of single parameter is extended to convert to the normal distribution and the optimal range determined according to normal distribution, or the initial range is reduced to seek the optimal range for calibration in the case of the limitation on range extension; for uniform distribution, the initial range is kept. In stage 3 the method of single parameter range selection (S-PRS) is performed on each parameter. Based on the indexes $S_E$ and $R_C$ estimated, the optimal combination of ranges is determined by coordinating the ranges selection for all parameters.

## 4. Results and discussion

### 4.1 Probability distribution characteristics of calibrated parameter values of Xinanjiang model

A series of calibrated parameters values were obtained through 100 times independent calibration by using GA method. The initial and calibrated ranges of parameter are presented in Table 3. The ratio of calibrated parameter range to initial one in Table 3 is less than 60% for most parameters (i.e. parameter CI, Kc, KI, SM, B, and WM), which implies that reducing the ranges can help calibrate the parameter efficiently. The 100 calibrated values for single parameters were normalized by dividing them by the corresponding initial range, and the box-plot chart of the results is shown in Fig. 5. It is obvious that the box and whiskers are symmetrical and the length of whiskers is longer than that of the box along the direction of Y axis for parameter CI, SM and Kc. But for other parameters, it is shown from the box-plot chart that the mean value deviates from the median one, which means a considerably asymmetric chart. According to these characteristics of the box-plot chart, it is indicated that the probability distribution of calibrated values is normal distribution for parameter CI, SM, and Kc, while that is exponential distribution for other parameters. The ratio of calibrated parameter range to initial one is less than 30% for parameters CI, SM, and Kc, while the ratio varies from 23% to 100% for parameters such as KI, B, CG, and Im. It suggest that





reducing the ranges is suitable to improve calibration for parameters whose values obey normal distributions, whereas that is not enough for parameters whose values obey exponential distributions.

## 4.2 Effect of range adjustment pattern on calibration results

Since the probability distribution of parameter value has a direct relation with parameter range selection, the range adjustment of parameters for calibration is discussed on the basis of probability distribution type of parameters in the investigation.

To normal distribution, reducing the range is generally used to select the appropriate range. Figure 6 shows the calibration results of parameter CI when the different parameter range are selected. The MINR (0.679–0.713) and MAXR (0.623–0.694) were picked out based on the cumulative frequency curve derived from calibration with initial range (0–0.900). From the cumulative curves and the histograms in Fig. 6a, 6b and 6c, it is found that the probability distribution of parameter CI values is converted from normal distribution to uniform distribution when the initial range is reduced to MINR, whereas the normal distribution is changed to the exponential one when the range is cut to MAXR. Figure 6d reveals that the contribution of parameter ranges selection to $E_{NS}$. It is found that the minimum $E_{NS}$ except extreme outliers rises convincingly and $E_{NS}$ concentrates at larger value zone when MINR is used instead of the initial range. It is indicated that the reduced range of high probability density is helpful to make calibration more steady and efficient.

To exponential distribution, both reduced range and the extended range of reasonable meaning can be used to select the appropriate range for calibration. Figure 7 shows the calibration results of parameter KI. Since the initial range of parameter KI can not be extended, the reduced range was searched by using the cumulative frequency curve, the MINR (0.660–0.700) and MAXR (0.522–0.660) were picked out. From the cumulative curves and the histograms in Fig. 7a, 7b and 7c, it is found that the probability distribution of parameter KI values is converted from exponential distribution to uniform distribution when the initial range is reduced to MINR, whereas the exponential distribution is still kept when the range is cut to MAXR. The contribution of parameter ranges to $E_{NS}$ is shown in Fig. 7d. Similar to the results of parameter CI, MINR is best for calibration when compared with MAXR or initial range. It is demonstrated that MINR is better than MAXR to improve calibration when reducing the range for parameters whose value obeys normal or exponential distribution. Because the parameter values in MINR indicate high probability to be pick out to achieve high $E_{NS}$, vice versa.

Figure 8 shows the calibration results of parameter B whose range can be extended. The initial range (B=0.1–0.4) of parameter B is common for most areas, but it is quite different for karst areas where the soil moisture storage varies remarkably with space, and as a result, the value of parameter B could be larger than 0.4. From Fig. 8a and 8b, it is shown that the distribution of parameter B is converted from exponential distribution to normal distribution when the initial range is extended to new one (B=0.1–0.6). After MINR selection is performed on the initial range and the extended range respectively, the two ranges, i.e. MINR (B=0.36–0.40) and extension-MINR (B=0.379–0.488) are obtained and then used to calibrate parameter B. From Fig. 8c and 8d, it is found that the probability distribution of parameter B values is converted into approximate uniform distribution when the range is reduced from initial range to MINR or from the extended range to extension-MINR. The box-plot chart of $E_{NS}$ for different ranges are shown in Fig. 8e. It is indicated that there is a considerable improvement of both



maximum and minimum $E_{NS}$ when extension-MINR is used for calibration. It suggests that an appropriate range extension followed by MINR selection is helpful to improve calibration for parameters whose probability distribution is exponential and ranges can be extended.

### 4.3 Effect of multiple parameters ranges combination on calibration results

The S-PRS method was employed to select the one-parameter optimal range for each parameter, and the optimal ranges, indexed $R_C$ and $S_E$ values are listed in Table 4. It is obvious that $R_C$ value in columns of parameter CI and WM are positive, but most $R_C$ values in column of parameter Im are negative. The negative $R_C$ value between two parameters indicates that using the optimal range of one parameter is adverse to calibration of the other parameter. Specially, both $R_{C\ EX,Im}$ and $R_{C\ Im,EX}$ are negative in spite of small values. It means that using the optimal ranges of parameter EX and Im simultaneously is not

conductive to multi parameter combined calibration. The mean of $R_C$ ($R_{C\ mean}$) varies with parameters. Parameter CI has the maximum $R_{C\ mean}$ of 0.465, while parameter Im the minimum $R_{C\ mean}$ of –0.026. Furthermore, $R_{C\ mean}$ values for all parameters are positive except for that for parameter Im. It is due to the accumulative negative influence of parameter Im on others.

To coordinate the contradiction between parameters, the index $S_E$ is used to pick parameters of high sensitivity to $E_{NS}$. From Table 4, it is found that parameter CI has the maximum $S_E$ of 54.7%, and parameter Im the minimum $S_E$ of 0.3%. Most $S_E$

values are more than 20% except those of parameters C, EX and Im. It suggests that parameters CI, B,  SM, KI, $K_c$, WM and CG are of high sensitive to $E_{NS}$, and parameters C, EX and Im of low sensitivity for $E_{NS}$. CI is the most sensitive parameter while Im the most insensitive parameter, which agrees with the work of Lü et al. (2013) and Song et al. (2013). For the well-developed karst areas, the thin layer of soil and strong permeability of limestone make rainfall easy to penetrate into the ground. Moreover, the existence of karst caves and subsurface streams contribute to great interflow storage which accounts for a large

proportion of streamflow. As a result, parameter KI representing penetrate ability of free water to interflow, and parameter CI representing recession capacity of interflow storage have significant influence on rainfall-runoff simulation results. Hence, parameters KI and CI are very sensitive in the investigation. It can be deduced that the optimal range of insensitive parameter Im can not be taken into account when there is contradiction owing to it, in order to improve calibration.

In order to determine the optimal range combination of multi parameter, seven cases are investigated with different range

combinations pf parameters (Table 5). The results of seven cases are compared in Fig. 9. There is a little decrease in $E_{NS}$ when Case 4 compared with Case 1, Case 2 and Case 3 It can be explained that both $R_{C\ EX,Im}$ and $R_{C\ Im,EX}$ are negative and the combination of the optimal ranges corresponding to the two parameters leads to a worse result. As $S_E$ of parameter Im is less than that of parameter EX, parameter EX is given priority to select the optimal range, that is why the calibration result of Case 3 is better than that of Case 2. As for the cases of multi-parameter range selection (i.e. Case 5, Case 6 and Case 7), the results

are much better than that of Case 1–4. There are some differences in $E_{NS}$ between Case 5, Case 6 and Case 7 when their box-plot charts are magnified. The box and whisker of $E_{NS}$ for Case 6 rise, which means Case 6 has a better performance of calibration than Case 5 does, when the optimal range of parameter CG is included. But the box and whisker of $E_{NS}$ for Case 7



decline when the optimal range of parameter Im is included. Because the mean $R_C$ value of parameter Im is negative and its $S_E$ much less than that of others, using the optimal range of Im is adverse to multi-parameter combined calibration.

## 5. Conclusions

Considering that there is the relation between the parameter ranges and probability distributions of parameter value, an approach to determine the optimal range combination for multi parameters of hydrological models is put forward by analysing the parameter value probability distribution, parameter sensitivity and parameter correlation. A case of improving the calibration of the GA-based Xinanjiang model for karst areas is studied, and some findings are presented as follows:

The proposed parameter range selection (PRS) method improves the minimum $E_{NS}$ and the maximum $E_{NS}$, which makes the results concentrate at high $E_{NS}$. The PRS-based calibration is, therefore, more efficient and effective. In the Xinanjiang model for karst areas, the parameters CI, Kc, SM and B obey normal probability distribution, and parameters WM, C, EX, KI, CG and Im obey exponential probability distributions. For normal distribution, the minimum ranges (MINR) with a given cumulative frequency of the parameter is preferred to be selected as the optimal range for calibration. For exponential distribution, if the parameter range can be extended outside the boundary of high probability, the extension range followed by MINR is recommended to be selected, otherwise MINR of the initial range is selected as the optimal range for calibration.

The $R_C$ and $S_E$ are two important indexes to help coordinate ranges of different parameter to form the optimal combination of ranges. As far as Xinanjiang model is concerned, the initial ranges of parameter of low $S_E$ and negative $R_{C\ mean}$ are preferred to adopted, and the optimal ranges of parameters of positive $R_{C\ mean}$ are used for multi parameter combined calibration.

**Acknowledgements.** The investigation is financially supported by special funds for scientific research on public causes of Chinese Ministry of Water Resources (No. 201401057) and the Scientific Research Foundation for the Returned Overseas Chinese Scholars, State Education Ministry (No. 2013–1792).

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





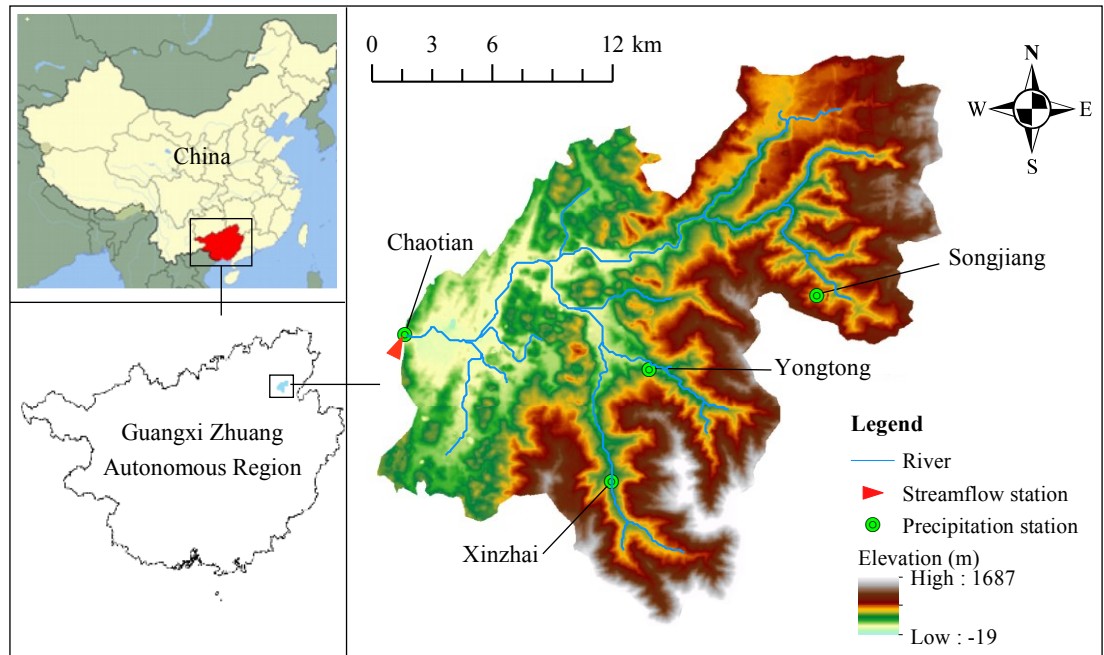

Fig. 1. Location of the study area




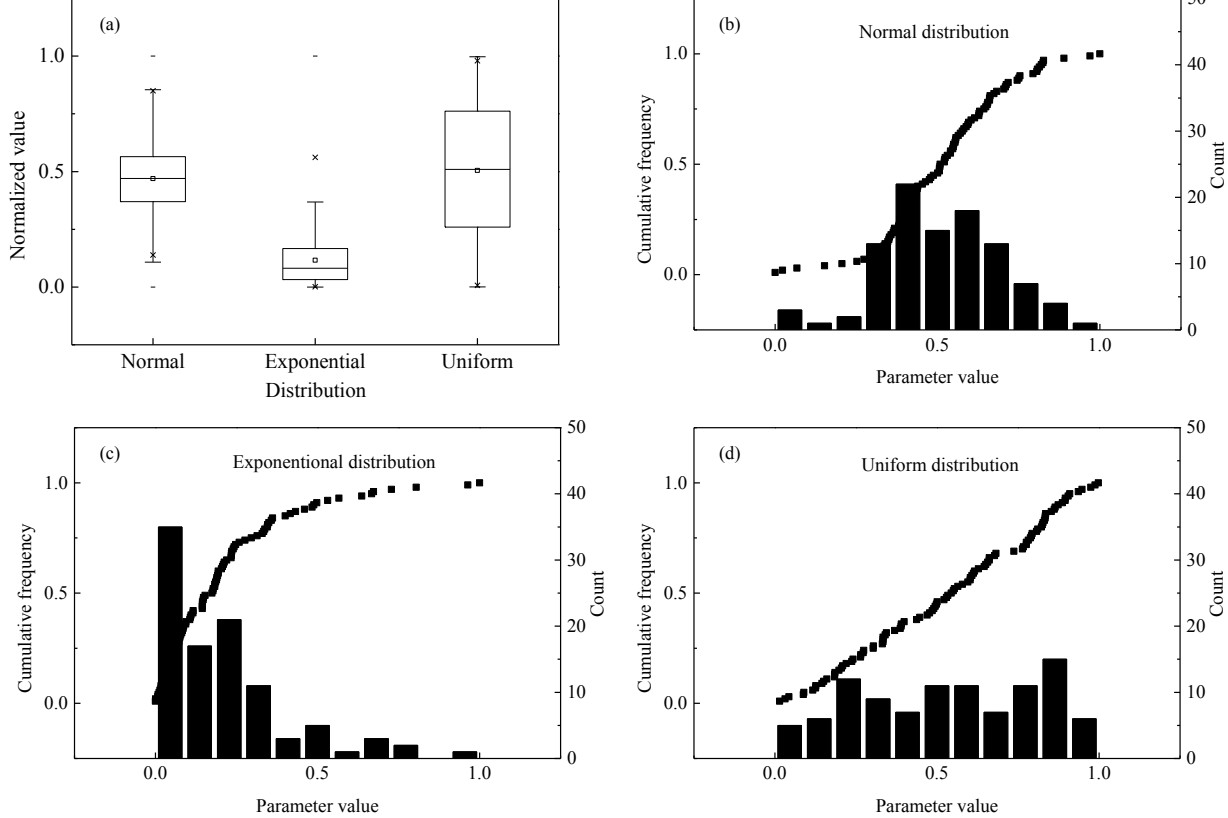

Fig. 2. Different probability distribution types of calibrated parameter values

(a) Box-plot charts of normal, exponential and uniform distribution (b) Cumulative frequency cure and histogram of normal distribution

(c) Cumulative frequency cure and histogram of exponential distribution (d) Cumulative frequency cure and histogram of uniform

5                                                                          distribution

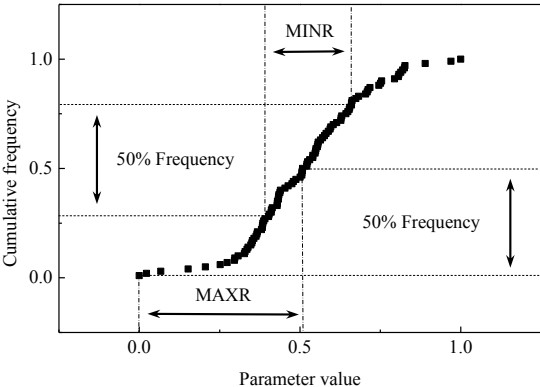

Fig. 3. Selection of minimum and maximum range (MINR and MAXR) with a cumulative frequency of 50%







Fig. 4. Flow chart of multiple parameters ranges selections





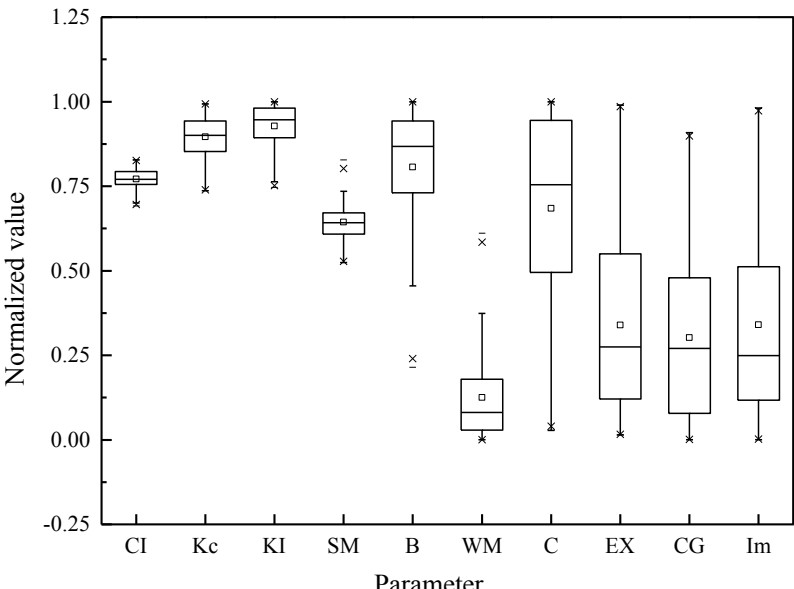

Fig. 5. The box-plot chart of normalized calibrated values for parameters of Xinanjiang model





Fig. 6. Results of range selection of parameter CI

(a) probability distribution of parameter values for schema initial range (b) probability distribution of parameter value for schema CI-MINR

(c) probability distribution of parameter values for schema CI-MAXR (d) box-plot chart of $E_{NS}$ for three schemas





Fig. 7. Results of range selection of parameter KI

(a) probability distribution of parameter values for schema initial range (b) probability distribution of parameter values for schema KI-MINR

(c) probability distribution of parameter values for schema KI-MAXR (d) box-plot chart of $E_{NS}$ for three schemas





Fig. 8. Results of range selection of parameter B

(a) probability distribution for schema initial range (b) probability distribution for schema B–Extension (c) probability distribution for

schema B–MINR (d) probability distribution for schema B–Extension–MINR (e) box–plot chart of $E_{NS}$ for four schemas





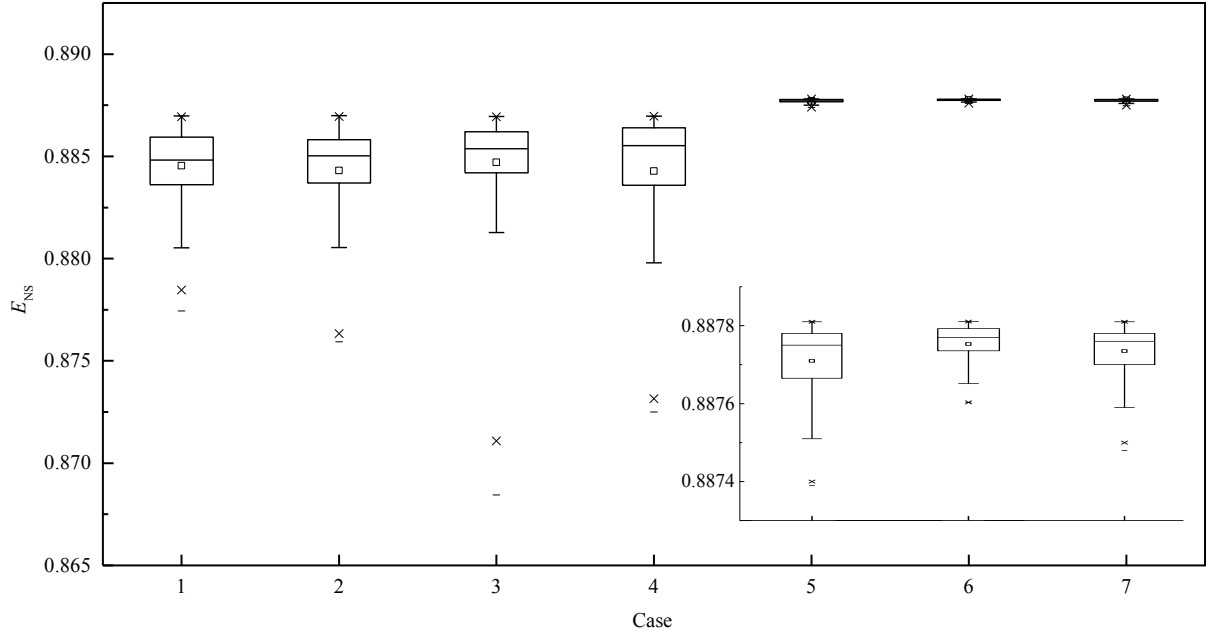

Fig. 9. Box–plot chart of $E_{NS}$ for different cases



Table 1. Metro-hydrological statistical data

| Year | $Q_{Max}$ (m³/d) | $Q_{Min}$ (m³/d) | $Q_{Avg}$ (m³/d) | $P_{Max}$ (mm/d) |
|------|------|------|------|------|
| 1996 | 719 | 0.76 | 14.38 | 235 |
| 1997 | 308 | 0.76 | 14.32 | 155 |
| 1998 | 369 | 0.66 | 13.67 | 157 |
| 1999 | 282 | 0.53 | 12.81 | 144 |
| 2000 | 339 | 1.14 | 11.37 | 107 |

$Q$ means streamflow and $P$ means average precipitation.

Table 2. Parameters of Xinanjiang model

| Parameter | Definition | Range |
|-----------|------------|-------|
| Kc | Ratio of potential evapotranspiration to pan evaporation | 0–1.1 |
| C | Coefficient of the deep layer, that depends on the proportion of the basin area covered by vegetation with deep roots | 0.1–0.2 |
| WM | Averaged soil moisture storage capacity of the whole layer | 120–200 (mm) |
| B | Exponential parameter with a single parabolic curve, which represents the non-uniformity of the spatial | 0.1–0.4 |
| Im | Percentage of impervious and saturated areas in the catchment | 0.01–0.04 |
| SM | Areal mean free water capacity of the surface soil layer, which represents the maximum possible deficit of free water storage | 10–50 (mm) |
| EX | Exponent of the free water capacity curve influencing the development of the saturated area | 1.0–1.5 |
| KI | Outflow coefficients of the free water storage to interflow | 0–0.7 |
| KG | relationships Outflow coefficients of the free water storage to groundwater relationships | KG+KI=0.7 |
| CG | Recession constants of the groundwater storage | 0.950–0.998 |
| CI | Recession constants of the lower interflow storage | 0–0.9 |



Table 3. Range changes of parameters in schema Initial

| Parameter | Initial parameter range | Calibrated parameter range* | Ratio** (%) |
|---|---|---|---|
| CI | 0–0.9 | 0.630–0.745 | 12.78 |
| Kc | 0–1.1 | 0.81–1.09 | 25.45 |
| KI | 0–0.7 | 0.534–0.7 | 23.71 |
| SM | 10–50 | 31–39.4 | 21.00 |
| B | 0.1–0.4 | 0.238–0.4 | 54.00 |
| WM | 120–200 | 120–150 | 37.50 |
| C | 0.1–0.2 | 0.1–0.2 | 100.00 |
| EX | 1.0–1.5 | 1.0–1.5 | 100.00 |
| CG | 0.950–0.998 | 0.950–0.994 | 91.67 |
| Im | 0.01–0.04 | 0.01–0.04 | 100.00 |

* the calibrated parameter range except the extreme outlier

** the ratio is the ratio of calibrated parameter range to initial parameter range

5 Table 4. The indexed $R_C$ and $S_E$ of parameters when optimal range for single parameter is performed for calibration

| Parameter* | CI | Kc | KI | SM | B | WM | C | EX | CG | Im |
|---|---|---|---|---|---|---|---|---|---|---|
| Optimal range of single parameter | 0.679–0.713 | 0.95–1.05 | 0.66–0.7 | 35–39 | 0.379–0.488 | 105–110 | 0.175–0.2 | 1–1.118 | 0.95–0.966 | 0.01–0.0245 |
| $R_C$ CI | 1.000 | 0.334 | 0.371 | 0.462 | 0.322 | 0.113 | 0.105 | 0.115 | –0.128 | 0.272 |
| Kc | 0.689 | 1.000 | 0.467 | 0.429 | 0.504 | 0.503 | 0.389 | 0.102 | 0.284 | 0.150 |
| KI | 0.778 | 0.315 | 1.000 | 0.445 | 0.574 | 0.268 | 0.456 | 0.328 | 0.060 | 0.258 |
| SM | 0.508 | –0.199 | 0.422 | 1.000 | –0.089 | 0.009 | –0.063 | 0.383 | 0.218 | –0.032 |
| B | 0.914 | 0.560 | 0.698 | –0.017 | 1.000 | 0.972 | –0.175 | 0.007 | –0.319 | –0.722 |
| WM | 0.575 | 0.311 | 0.439 | 0.553 | 0.325 | 1.000 | 0.229 | 0.360 | –0.069 | –0.235 |
| C | 0.208 | 0.273 | 0.083 | 0.151 | 0.277 | 0.335 | 1.000 | 0.077 | 0.200 | 0.210 |
| EX | 0.054 | 0.047 | –0.011 | 0.018 | 0.371 | 0.045 | 0.009 | 1.000 | –0.021 | –0.025 |
| CG | 0.221 | 0.246 | –0.135 | 0.022 | 0.010 | 0.198 | –0.034 | –0.009 | 1.000 | –0.112 |
| Im | 0.238 | 0.073 | –0.025 | 0.045 | 0.031 | 0.030 | –0.026 | –0.020 | 0.001 | 1.000 |
| Mean of $R_C$ | 0.465 | 0.218 | 0.257 | 0.234 | 0.258 | 0.275 | 0.099 | 0.149 | 0.025 | –0.026 |
| $S_E$ (%) | 54.7 | 47.9 | 36.6 | 41.7 | 48.1 | 39.9 | 10.8 | 14.7 | 21.9 | 0.3 |

* The parameter represent the parameter X in Eq. 2.



Table 5. Parameter ranges setting for different cases

| Case | Range setting of parameter | | | | | | | | | |
|------|-----|-----|-----|-----|-----|-----|-----|-----|-----|-----|
|      | WM  | C   | B   | SM  | EX  | KI  | CI  | CG  | Kc  | Im  |
| 1    | I   | I   | I   | I   | I   | I   | I   | I   | I   | I   |
| 2    | I   | I   | I   | I   | I   | I   | I   | I   | I   | O   |
| 3    | I   | I   | I   | I   | O   | I   | I   | I   | I   | I   |
| 4    | I   | I   | I   | I   | O   | I   | I   | I   | I   | O   |
| 5    | O   | O   | O   | O   | O   | O   | O   | I   | O   | I   |
| 6    | O   | O   | O   | O   | O   | O   | O   | O   | O   | I   |
| 7    | O   | O   | O   | O   | O   | O   | O   | O   | O   | O   |

The symbol 'I' represents the initial range of the parameter in Table 3, and 'O' the optimal range of the parameter in Table 4.