# Peer review of "Improvement of hydrological model calibration by selecting multiple parameter ranges"

_Hydrology and Earth System Sciences, 2016_

## Referee Comment (RC1) · Anonymous Referee #1 · 4 Aug 2016

Overview

\_\_\_\_\_

This study touches upon the problematic of obtaining inadequate calibration results in hydrological modelling as a result of inappropriate parameter ranges. To address this issue the authors present a new approach to select parameter ranges based on probability distribution characteristics of the parameter space. The methodology is divided in three main steps: (i) determination of the probability distribution of the calibrated parameter values, (ii) adjustment of the initial parameter ranges of each parameter individually by finding the ranges of maximum and minimum probability density for a given cumulative frequency, and (iii) resolving the impact of the adjusted parameter range on other parameters and keeping the modified range only if it has a positive

impact on the calibration of the other parameters.

In order to demonstrate the application of the proposed methodology the authors calibrated the Xinanjiang model for the Chaotianhe River catchment. Out of 10 model parameters that needed to be calibrated, the authors found that 3 parameters follow a normal distribution and the other 7 an exponential distribution. They then proceed to constrain the parameter ranges following the proposed approaches for each of the distributions and test a number of cases involving different combinations of parameters kept at the initial ranges and having optimized ranges. Finally, the authors conclude that the proposed approach make model calibration more effective.

General comments

——————-

This manuscript presents a new methodology to determine adequate parameter ranges in order to improve calibration results in hydrological modelling. The topic is suitable and interesting for the Hydrology and Earth System Sciences readership. I have particular comments on the interpretation of the results, the magnitude of the impact of the proposed technique on the model calibration results, and the language. I think that the manuscript has a potential to make a good contribution to the subject area of hydrological modelling and I recommend that it should be accepted after moderate revisions based on the comments listed below.

Main comments

————-

1) Probability distribution functions considered in the study. In this study, the authors propose a method to constrain parameter ranges for parameters that follow uniform, normal, and exponential probability distribution functions. These are the probability distribution functions that the case study model parameters reportedly follow. Some of the claims are debatable. For instance, parameters CI and Kc are reported to follow

normal distributions (page 7, line 29) based on the following statement (page 7, line 25): "It is obvious that the box and whiskers are symmetrical and the length of whiskers is longer than that of the box [...].". Looking at Fig. 5, however, the whiskers are not symmetrical and, on the upper side, not longer than the box, suggesting that the ranges of these parameters do not follow a normal probability distribution. Therefore, the method used to constrain the ranges of these parameters might not be the optimal, potentially changing the results of the study.

2) The authors report that parameter range selection has a direct impact on calibration efficiency and propose a new method to improve model calibration (page 1, line 12). The reported results, however, indicate that the improvement in the calibration efficiency by the proposed methodology is quite modest. For instance, in Fig. 9 different cases involving different combinations of parameters keeping the initial range and others having the "optimal" range are compared. The model efficiency different between case I (all the parameters set at the initial ranges) and any other of the considered cases is of the order of 0.002 at best. This suggests that the benefits of using the proposed technique are small.

3) The language should be improved to make the manuscript easier to understand and more compelling. More specifically, the following aspects should be revised: verb tenses (e.g. page 3, line 23-24: "single parameter is selected" - "correlation and sensitivity were estimated"; page 6, line 15: "The index Rc was quantified" instead of "The index Rc were quantified"), spelling errors (e.g. page 6, line 13: "contribute" instead of "contributes"; page 9, line 25: "of" instead of "pf"), and sentence structure (e.g. page 9, line 15 "[...] parameters [...] are of high sensitive to Ens"). I would strongly recommend the article to be checked for language.

Specific comments
* * *
1) page 4, line 28 and page 6, line 2: "plenty of tests". The text suggests that the

authors defined their sampling size and cumulative frequency value through a process of trial and error. Since this might affect the subsequent results I think that evidence should be provided to support the authors' claims.

2) Page 8, line 34: "[...] there is considerable improvement [...]". "Considerable" is a vague word, please provide a quantitative measure of the improvement. Similar problem in page 8, line 12. Please revise the results section to ensure that no vague words are used.

3) Page 9, line 24: Seven cases are investigated with different combinations of parameter ranges. What is the rationale behind the chosen combinations? Please specify.

4) Figure 1: The chosen color scale makes the figure difficult to read in black and white. Please consider modifying it to facilitate reading the figure in printed form. The elevation units should be "m a.s.l." instead of "m". The lowest elevation in the catchment is reported to be 19 m below the sea level; is that so?

5) Figure 2: Please correct "cure" in the figure caption.

6) Figure 5: Since the figure represents normalized parameter values on the y-axis, it would be more informative to constrain this axis between 0 and 1.

7) Table 2: please provide units for all the parameters. In the case of dimensionless parameters indicate so.

8) Table 2, 3, 4: In order to facilitate the readability of the different tables it might be convenient to order the parameters in the same way in all the tables.
* * *

---

## Referee Comment (RC2) · Anonymous Referee #2 · 5 Aug 2016

Overview and general comments:

In this study the Authors propose a new methodology for the calibration of hydrological model. The work intends to provide a general framework that can optimize the model calibration in case of multi-parameters that may have different value ranges, as well as different sensitivity and correlation behaviors. The analysis is performed referring to a specific case study, Chaotianhe river in the Southwest China, for which the Xinanjiang hydrological model, characterized by 10 parameters, has been implemented. The study reports the application of the proposed methodology showing an improvement on model performance.

In general, the manuscript is well organized and the methodology is sufficiently described. However, in many parts, the writing is not fluent and with grammatical errors.

In general I can say that the methodology sounds interesting and may be of interest for the hydrological community. Even though I have some doubts on the real effectiveness of the proposed approach (improvements are not really significant; see comment below), I think the paper can be considered for publication after a moderate revision.

Hereafter some general considerations and specific comments:

- My main concern is related to the real impact of the proposed methodology. The benefit in terms of NSE is very small: see Fig. 9. Is this improvement relevant for hydrological application? If we focus exclusively on model performances I do not think this methodology shows a significant improvement. I suggest to emphasize more the physical considerations that may rise from the application, for example in terms of sensitivity of specific parameters in relation to the particular nature of the study area, or regarding the evaluation of parameters correlation. From my point of view this methodology may provide additional insights regarding the interactions among model parameters under different hydrological conditions. In other words: since the improvement in terms of NSE seems to be not relevant, what are the added values of this methodology compare to existing ones?

- Continuing on the effectiveness of the methodology, the Author do not provide any information regarding the initial GA calibration. Are there benefits from the application of the methodology in terms of NSE values? What are the computational/time efforts required for the implementation of the calibration framework compared to other techniques?

- Is there a specific reason for considering the MAXR range interval in addition to MINR (see Figure 3). Why a modeler should consider the range of minimum probability density of the parameter values? If it is not necessary I suggest to consider its removal from the analysis.

- At P7, line 25. Why this is obvious? Looking at Fig. 5 this is not. Do the Authors apply statistical tests to evaluate the statistical distribution of the parameters?

- Concerning the 7 scenarios reported in table 5, how have you defined them? Are there specific reasons behind the use of initial or optimal ranges for cases 5, 6 and 7? In addition, I suggest to keep the same column order for parameters, it's easier to read table 5 in relation to the values of table 4.

- The writing in some part of the manuscript should be improved. I suggest to carefully go through the overall manuscript and check verbs and syntax (here some example: P5,L3; P5,L23; P6,L5, P9,L25-26; . . . ; P10,L17).

Specific comments:

- Abstract: in the last part of the abstract, roughly from line 20 on, the Authors report some specific methodological considerations that may not be really clear to one who has not already read the paper. I suggest to focus more on the scope and aims of the analysis, reporting also that the methodology proposes indexes for the evaluation of parameter sensitivity and correlations, as well as a summary of the main outcomes.

- P1, L29: is "method" appropriate to indicate hydrological process modelling? I would suggest something like "tools" or similar.

- P4, L28: On which base you say that 100 samples are enough? Have you adopted some statistical texts to verify the statistical distribution of the considered parameters.

- P4, L3: " A Genetic Algorithm (GA) was selected"

- P9, L6: why do you say that it is obvious?

- P10,L7: please remove the colon;

-Fig. 2: check "curve"; I also suggest to re-word the caption as: [. . .]; Cumulative frequency and [. . .] distribution for normal (b), exponential (c) and uniform (d) distributions.

-Fig. 6, 7 and 8: is it necessary to report the label "schema"?

- Table 1: is P the average or the max?

- Table 2: the definition of parameter B seems not complete. Also, the column "range" of Table 2 is reported twice (see Table 3).

- Table 3: the main legend is not really clear, I suggest to re-word it. ** "ratio of calibrated parameter . . ."

---

## Author Response (AR1)

**Part 1 Responses to Referee's comments**

We appreciate the comments from the reviewer and truly believe these comments can help us to improve our manuscript. We consider the corresponding changes can be included in the revised document to achieve publication status. We provide responses to the main and specific comments in sequential order as follow:

**Responses to Referee #1**

**Main comment #1**

"Probability distribution functions considered in the study. In this study, the authors propose a method to constrain parameter ranges for parameters that follow uniform, normal, and exponential probability distribution functions. These are the probability distribution functions that the case study model parameters reportedly follow. Some of the claims are debatable. For instance, parameters CI and Kc are reported to follow normal distributions (page 7, line 29) based on the following statement (page 7, line 25): "It is obvious that the box and whiskers are symmetrical and the length of whiskers is longer than that of the box [...].". Looking at Fig. 5, however, the whiskers are not symmetrical and, on the upper side, not longer than the box, suggesting that the ranges of these parameters do not follow a normal probability distribution. Therefore, the method used to constrain the ranges of these parameters might not be the optimal, potentially changing the results of the study."

**Responses to main comment #1:**

There are three types of distribution discussed in the investigation. In order to distinguish them, a simple method in section 3.2.2 was used based on shapes of the cumulative frequency curve and the histogram as well as the sizes of whiskers and box in the box-plot. Despite simplicity, it is subjective and unintelligible to readerships. For avoiding the confusion as described in this comment, a Kolmogorov-Smirnov (K-S) test will be employed to objectively identify each distribution type in the revised paper. Indeed, we carried out K-S tests to evaluate statistical distributions of all parameters in the hydrologic model. The results of K-S tests for parameters CI, Kc, and SM are listed in the following Table A. It is shown that both exponential and uniform distributions are rejected for the three parameters while normal distribution is not. It implies that the three parameters follow normal distributions. Therefore, the simple method used earlier does not change the results of the study, although it is subjective.

Table A. The results of K-S tests for parameters CI, Kc, and SM

| | CI | | | Kc | | | SM | | |
|---|---|---|---|---|---|---|---|---|---|
| | Normal | Exponential(2P) | Uniform | Normal | Exponential(2P) | Uniform | Normal | Exponential(2P) | Uniform |
| Statistic | 0.0623 | 0.32805 | 0.1151 | 0.09199 | 0.37961 | 0.10694 | 0.05983 | 0.30392 | 0.10982 |
| P-Value | 0.80925 | 5.40E-10 | 0.1306 | 0.34466 | 3.08E-13 | 0.18882 | 0.84521 | 1.23E-08 | 0.16628 |
| $\alpha$ | 0.2 | 0.01 | 0.2 | 0.2 | 0.01 | 0.2 | 0.2 | 0.01 | 0.2 |
| Reject? | No | Yes | Yes | No | Yes | Yes | No | Yes | Yes |

**Main comment #2**

"The authors report that parameter range selection has a direct impact on calibration efficiency and propose a new method to improve model calibration (page 1, line 12). The reported results, however, indicate that the improvement in the calibration efficiency by the proposed methodology is quite modest. For instance, in Fig. 9 different cases involving different combinations of parameters keeping the initial range and others having the "optimal" range are compared. The model efficiency different between case I (all the parameters set at the initial ranges) and any other of the considered cases is of the order of 0.002 at best. This suggests that the benefits of using the proposed technique are small."

**Responses to main comment #2:**

Notwithstanding a small increase in maximum $E_{NS}$, there is a significant improvement in minimum $E_{NS}$ by using the proposed method. Comparing case 6 (using the optimal combination of ranges) with case 1 (using the initial ranges) in Fig. 9, we find that the maximum $E_{NS}$ increases by 0.001 while the minimum $E_{NS}$ (except outliers) increases by 0.01. The rising minimum $E_{NS}$ with the fixed maximum contributes to the shrinkage of the range of the possible solutions. As a result, the uncertainty of the model performance can be effectively controlled. Moreover, the methodology can be used to analyze the parameter correlation and sensitivity by computing two indexes $R_{C Y,X}$ and $S_E$. The paper presents the preliminary study of the proposed methodology. In the preliminary study, we adopt a Xinanjiang model with several parameters to evaluate the calibration efficiency of the methodology. Since the parameter Im having negative effect on other parameters is a little bit insensitive in a Xinanjiang model, a modest improvement in calibration efficiency is found after the application of the methodology. In future, we will consider using other complicated hydrologic models with more parameters to further study the application of the methodology.

**Main comment #3**

"The language should be improved to make the manuscript easier to understand and more compelling. More specifically, the following aspects should be revised: verb tenses (e.g. page 3, line 23-24: "single parameter is selected" - "correlation and sensitivity were estimated"; page 6, line 15: "The index $R$c was quantified" instead of "The index $R$c were quantified"), spelling errors (e.g. page 6, line 13: "contribute" instead of "contributes"; page 9, line 25: "of" instead of "pf"), and sentence structure (e.g. page 9, line 15 "[...] parameters [...] are of high sensitive to $E_{ns}$"). I would strongly recommend the article to be checked for language."

**Responses to main comment #3:**

We will revise the manuscript as the suggestion:

Page 3, line 23-24: "single parameter is selected" >> "single parameter was selected"

Page 6, line 15: "The index $R$c were quantified" >> "The index $R$c was quantified"

Page 6, line 13: "contributes" >> "contribute"

Page 9, line 25: "pf" >> "of"

Page 9, line25-26: "...when Case 4 compared with Case 1, Case 2 and Case 3 It can be explained..." >> "... when Case 4 is compared with Case 1, Case 2 and Case 3. It can be explained..."

Page 9, line 15: "… and CG are of high sensitive to $E_{NS}$"   >> "... and CG are highly sensitive to

$E_{NS}$"

In addition, we will check the paper carefully and correct the other language errors. For example:

Page 1, line 3: "Qiaofeng." >> "Qiaofeng"

Page 1, line 14: "characteristics of single parameter value was analysed" >> "of single parameter value was analysed"

Page 1, line 17: "corresponding to the distribution" >> "corresponding to the distribution type"

Page 2, line 4: "mechanism of water cycle" >> "mechanism of the water cycle"

Page 2, line 9: "the streamflow at catchment outlet" >> "the streamflow at the catchment outlet"

Page 3, line 30: "in flood reason" >> "in flood season"

Page 4, line 18: "from observed streamflow" >> "from the observed streamflow"

Page 5, line 3: "which representing agreement between observed and simulated data" >> "which represents the agreement between observed and simulated data"

Page 5, line 11: "whisker to the box" >> "the whisker to the box"

Page 5, line 23: "…, the initial range of parameter is required adjusting properly" >> "…, the initial range of parameter requires adjusting properly"

Page 5, line 28: "represents the ranges" >> "represent the ranges"

Page 6, line 4: "in case of larger percentage" >> "in case of a larger percentage"

Page 6, line 5: "... values can transform and finally convert into..." >> "... values can be converted into..."

Page 6, line 11: "may more and less effect" >> "may affect"

Page 6, line 17: "the greater positive influence" >> "greater positive influence"

Page 7, line 2: "The statistic analysis" >> "The statistical analysis"

Page 7, line 6: "ranges is substituted" >> "ranges are substituted"

Page 7, line 8: "the selected one is adopted for calibration of multiple parameters." >> "the selected ones are adopted for multi-parameters model calibration."

Page 7, line 16: "In stage 3 the ... " >> "In stage 3, the ..."

Page 7, line 26: "direction of Y axis" >> "direction of the Y axis"

Page 7, line 30-31: "The ratio of calibrated parameter range to initial one is less than 30% for parameters CI, SM, and Kc" >> "The ratios of the calibrated parameter range to the initial one are less than 30% of parameters CI, SM, and Kc"

Page 7, line 31: "It suggest that" >> "It suggests that"

Page 8, line 6: "To normal distribution" >> "For normal distribution"

Page 8, line 7: "different parameter range are selected" >> "different parameter ranges are selected"

Page 8, line 13: "concentrates at larger value zone" >> "concentrates at a higher value range"

Page 8, lien 24: "Because the parameter values in MINR indicate a high probability to be picked out to achieve high $E_{NS}$, vice versa." >> "It is because that the parameter values that may achieve a higher $E_{NS}$ can be easily picked out from the MINR of higher probability density."

Page 8, line 33: "box-plot chart of $E_{NS}$ for different ranges are shown in Fig. 8e" >> "box-plots for different ranges are shown in Fig. 8e"

Page 9, line 6: "value in columns" >> "values in columns"

Page 9, line 20: "penetrate" >> "penetrability"

Page 9, line 23: "there is contradiction owing to it" >> "there is a contradiction owing to it"

Page 10, line 13: "the extension range followed by" >> "the extended range followed by"

Page 10, line 17: "to adopted" >> "to be adopted"

**5   Specific comment #1**

"Page 4, line 28 and page 6, line 2: "plenty of tests". The text suggests that the authors defined their sampling size and cumulative frequency value through a process of trial and error. Since this might affect the subsequent results I think that evidence should be provided to support the authors' claims."

**Responses to specific comment #1:**

10   Before defining sampling size and cumulative frequency value, we performed a lot of trial tests. Figure A shows the variation curves of maximum and minimum $E_{NS}$ with sample size. It is indicated that both maximum and minimum $E_{NS}$ keep stable when sampling size is greater than 100. Avoiding the time-consuming computation, we assigned sampling size for the study as 100. Figure B gives the variation curves of maximum and minimum $E_{NS}$ with cumulative frequency value. It is found that the maximum

15   $E_{NS}$ keeps constant despite a cumulative frequency value varying, while the minimum $E_{NS}$ approaches the peak value of 0.881 when the cumulative frequency value is equal to 50%. Considering that higher minimum $E_{NS}$ contributes to more efficient calibration, we selected the fixed cumulative frequency value of 50% to determine the ranges of maximum and minimum probability density (i.e. MINR and MAXR) for each parameter.

[Figure]

Figure A. Variation curves of maximum and minimum $E_{NS}$ with sample size

[Figure]

Figure B. Variation curves of maximum and minimum $E_{NS}$ with cumulative frequency value

**Specific comment #2**

"Page 8, line 34: "[...] there is considerable improvement [...]". "Considerable" is a vague word, please provide a quantitative measure of the improvement. Similar problem in page 8, line 12. Please revise the results section to ensure that no vague words are used."

**Responses to specific comment #2:**

According to the review comments, we will revise the corresponding parts as follows:

Page 8, line 34: "It is indicated that there is a considerable improvement of both maximum and minimum $E_{NS}$ when extension-MINR is used for calibration."  >> "It is shown from Fig. 8e that there is little improvement in maximum $E_{NS}$ when MINR is used for calibration instead of the initial range. There is an increase of 0.0003 in maximum $E_{NS}$ if the initial range is replaced with the extension range or extension-MINR. As for minimum $E_{NS}$ (except outliers), an increase of 0.001 in the case of MINR, a decrease of 0.003 in case of the extension range and an increase of 0.003 in the case of extension-MINR are found when the initial range is substituted with the three ranges respectively."

Page 8, line 12: "It is found that the minimum $E_{NS}$ except extreme outliers rises convincingly and $E_{NS}$ concentrates at larger value zone when MINR is used instead of the initial range." >> "It is found that the minimum $E_{NS}$ except extreme outliers rises from 0.8805 to 0.8842 and $E_{NS}$ concentrates at larger value zone when MINR is used instead of the initial range."

Page 9, line 29-30: "As for the cases of multi-parameter range selection (i.e. Case 5, Case 6 and Case 7), the results are much better than of Case 1-4."  >> "As for the cases with multi-parameter range selection (i.e. Cases 5-7), the results are much better than those of cases with initial range or single-parameter range selections (i.e. Cases 1-4). There is approximately an increase of 0.001 in maximum $E_{NS}$ and an increase of 0.01 in minimum $E_{NS}$ when the multi-parameter range selection is performed. "

**Specific comment #3**

"Page 9, line 24: Seven cases are investigated with different combinations of parameter ranges. What

is the rationale behind the chosen combinations? Please specify."

**Responses to specific comment #3:**

The seven cases were set to demonstrate three primary results. Firstly, the multi-parameter optimal range selection method is superior to the single-parameter one for calibrating hydrologic models with multiple parameters. It can be deduced from higher $E_{NS}$ values of Cases 5-7 than those of Cases 1-4. Secondly, merely using the optimal range of the parameter of relatively higher sensitivity contributes to more efficient calibration when the two parameters have negative effect on each other. It can be concluded by comparing the $E_{NS}$ values of Cases 2-4 referring to the two parameters EX and Im. Thirdly, the combination of optimal ranges of all parameters is not the optimum inasmuch as some parameters like Im have negative effects on other parameters. It can be inferred through analyzing the $E_{NS}$ values of Cases 5-7. The analysis of sensitivity and correlation between parameters is, therefore, very important to determine the optimum ranges combination of all parameters for model calibration.

**Specific comment #4**

"Figure 1: The chosen color scale makes the figure difficult to read in black and white. Please consider modifying it to facilitate reading the figure in printed form. The elevation units should be "m a.s.l." instead of "m". The lowest elevation in the catchment is reported to be 19 m below the sea level; is that so?"

**Responses to specific comment #4:**

We will use the gray ribbon for DEM rendering to make Figure 1 easy to read in printed form. The unit "m a.s.l." will be used instead of "m" in revised Figure 1. In addition, there exist dolines (known as sinkholes) in the catchment. It is the reason why the lowest elevation in the catchment is 19 m below the sea level.

**Specific comment #5**

"Figure 2: Please correct "cure" in the figure caption."

**Responses to specific comment #5:**

We will change "cure" to "curve" in the caption of Figure 2.

**Specific comment #6**

"Figure 5: Since the figure represents normalized parameter values on the y-axis, it would be more informative to constrain this axis between 0 and 1."

**Responses to specific comment #6:**

We will constrain the y-axis of Figure 5 between 0 and 1 in the revised paper. The Fig. 5 modified is presented as follows.

[Figure]

Fig. 5. The box-plot chart of normalized calibrated values for parameters of Xinanjiang model

**Specific comment #7**

"Table 2: please provide units for all the parameters. In the case of dimensionless parameters indicate so."

**Responses to specific comment #7:**

We will give units for parameters in a Xinanjiang model, as it is shown in Table 2 below.

**Specific comment #8**

"Table 2, 3, 4: In order to facilitate the readability of the different tables it might be convenient to order the parameters in the same way in all the tables."

**Responses to specific comment #8:**

We will modify Tables 2, 5 so that the parameters are ordered in the same way in the related tables. Moreover, the column "range" of Table 2 will be changed as column "Units" because the ranges for parameters are reported in Table 3.

Table 2. Parameters of Xinanjiang model

| Parameter | Definition | Units |
|---|---|---|
| CI | Recession constants of the lower interflow storage | dimensionless |
| Kc | Ratio of potential evapotranspiration to pan evaporation | dimensionless |
| KI | Outflow coefficients of the free water storage to interflow | dimensionless |
| SM | Areal mean free water capacity of the surface soil layer, which represents the maximum possible deficit of free water storage | mm |
| B | Exponential parameter with a single parabolic curve, which represents the non-uniformity of the spatial | dimensionless |
| WM | Averaged soil moisture storage capacity of the whole layer | mm |
| C | Coefficient of the deep layer, that depends on the proportion of the basin area covered by | dimensionless |

| EX | Exponent of the free water capacity curve influencing the development of the saturated area | dimensionless |
|---|---|---|
| CG | Recession constants of the groundwater storage relationships | dimensionless |
| KG* | Outflow coefficients of the free water storage to groundwater relationships | dimensionless |
| Im | Percentage of impervious and saturated areas in the catchment | dimensionless |

* the value of KG is calculated by the function 0.7-KI

Table 5. Parameter ranges setting for different cases

| Case | Range setting of parameter | | | | | | | | | |
|---|---|---|---|---|---|---|---|---|---|---|
| | CI | Kc | KI | SM | B | WM | C | EX | CG | Im |
| 1 | I | I | I | I | I | I | I | I | I | I |
| 2 | I | I | I | I | I | I | I | I | I | O |
| 3 | I | I | I | I | I | I | I | O | I | I |
| 4 | I | I | I | I | I | I | I | O | I | O |
| 5 | O | O | O | O | O | O | O | O | I | I |
| 6 | O | O | O | O | O | O | O | O | O | I |
| 7 | O | O | O | O | O | O | O | O | O | O |

The symbol 'I' represents the initial range of the parameter in Table 3, and 'O' the optimal range of the parameter in Table 4.

**Responses to Referee #2**

**Main comment #1**

"My main concern is related to the real impact of the proposed methodology. The benefit in terms of NSE is very small: see Fig. 9. Is this improvement relevant for hydrological application? If we focus exclusively on model performances I do not think this methodology shows a significant improvement. I suggest to emphasize more the physical considerations that may rise from the application, for example in terms of sensitivity of specific parameters in relation to the particular nature of the study area, or regarding the evaluation of parameters correlation. From my point of view this methodology may provide additional insights regarding the interactions among model parameters under different hydrological conditions. In other words: since the improvement in terms of NSE seems to be not relevant, what are the added values of this methodology compare to existing ones?"

**Responses to main comment #1:**

Notwithstanding a small increase in maximum $E_{NS}$, there is a significant improvement in minimum $E_{NS}$ by using the proposed method. Comparing case 6 (using the optimal combination of ranges) with case 1 (using the initial ranges) in Fig. 9, we find that the maximum $E_{NS}$ increases by 0.001 while the minimum $E_{NS}$ (except outliers) increases by 0.01. The rising minimum $E_{NS}$ with the fixed maximum contributes to the shrinkage of the range of the possible solutions. As a result, the uncertainty of the model performance can be effectively controlled. Moreover, the methodology can be used to analyze the parameter correlation and sensitivity by computing two indexes $R_{C\,Y,X}$ and $S_E$. The paper presents the preliminary study of the proposed methodology. In the preliminary study, we adopt a Xinanjiang model with several parameters to evaluate the calibration efficiency of the methodology. Since the parameter Im having negative effect on other parameters is a little bit insensitive in a Xinanjiang model, a modest improvement in calibration efficiency is found after the application of the methodology. In future, we will consider using other complicated hydrologic models with more parameters to further study the application of the methodology.

**Main comment #2**

"Continuing on the effectiveness of the methodology, the Author do not provide any information regarding the initial GA calibration. Are there benefits from the application of the methodology in terms of NSE values? What are the computational/time efforts required for the implementation of the calibration framework compared to other techniques?"

**Responses to main comment #2:**

Since the GA method is very common tool for parameter calibration of hydrologic models, we provide a little information about GA calibration. In the study, we carried out trial tests to determine the optimal combination of control parameters: crossover probability of 0.5, mutation probability of 0.7 for the individual, mutation probability of 0.5 for each gene, population size of 21, maximum generation number of 500 and maximum iteration number of 50. These parameters were kept constant for GA calibration in the investigation. The application of the proposed methodology results in an increase of 0.01 in minimum $E_{NS}$, compared with that of the pure GA method. The rising of minimum $E_{NS}$ with little change of the maximum may shrink the range of the possible solutions. As a result, the uncertainty of the model performance can be effectively controlled.

Through a run of calibration framework, a combination of values of all parameters and the corresponding $E_{NS}$ are obtained. Figure C shows the variation curves of maximum and minimum values of $E_{NS}$ with number of runs by using a GA method and a proposed PRS method, respectively. It is indicated from Figure A that no mater it is maximum or minimum $E_{NS}$, the value calculated with a proposed method is almost the same as that with a GA method when the number of runs is less than 100. If a proposed method is used for calibration instead of a GA method, there are approximately an increase of 0.001 in maximum $E_{NS}$ and an increase of 0.01 in minimum $E_{NS}$ when the number of runs is greater than 100. Thus, for any particular run number, the value of $E_{NS}$ calculated with a PRS method is not less than that with a GA method. The application of a proposed method, therefore, contributes to a more efficient calibration than that of a GA method does.

[Figure]

Figure C. the variation curves of maximum and minimum $E_{NS}$ with number of runs by using a GA method and a proposed PRS method

**Main comment #3**

"Is there a specific reason for considering the MAXR range interval in addition to MINR (see Figure 3). Why a modeler should consider the range of minimum probability density of the parameter values? If it is not necessary I suggest to consider its removal from the analysis."

**Responses to main comment #3:**

In order to figure out how the selections of two typical ranges, MINR and MAXR, affect respectively the calibration efficiency under different distribution types, we considered the MAXR range internal in Figure 3. From the results shown in Figures 6-7 (referring to parameter CI of a normal distribution and parameter KI of an exponential distribution), it is indicated that MINR is better than MAXR for improving calibration whichever distribution is specified. We removed, therefore, the MAXR range interval from the later analysis presented in Figure 8. As it is one of the main results of the study that MINR is better than MAXR for improving calibration, we would like to keep the MAXR range interval in Figures 3/6/7.

**Main comment #4**

"At P7, line 25. Why this is obvious? Looking at Fig. 5 this is not. Do the Authors apply statistical tests to evaluate the statistical distribution of the parameters?"

**Responses to main comment #4:**

There are three types of distribution discussed in the investigation. In order to distinguish them, a simple method in section 3.2.2 was used based on shapes of the cumulative frequency curve and the histogram as well as the sizes of whiskers and box in the box-plot. Despite simplicity, it is subjective and unintelligible to readerships. For avoiding the confusion as described in this comment, a Kolmogorov-Smirnov (K-S) test will be employed to objectively identify each distribution type in the revised paper. Indeed, we carried out K-S tests to evaluate statistical distributions of all parameters in the hydrologic model. The results of K-S tests for parameters CI, Kc, and SM are listed in the following Table B. It is shown that both exponential and uniform distributions are rejected for the three parameters while normal distribution is not. It implies that the three parameters follow normal distributions. Therefore, the simple method used earlier does not change the results of the study, although it is subjective.

Table B. The results of K-S tests for parameters CI, Kc, and SM

| | CI | | | Kc | | | SM | | |
|---|---|---|---|---|---|---|---|---|---|
| | Normal | Exponential(2P) | Uniform | Normal | Exponential(2P) | Uniform | Normal | Exponential(2P) | Uniform |
| Statistic | 0.0623 | 0.32805 | 0.1151 | 0.09199 | 0.37961 | 0.10694 | 0.05983 | 0.30392 | 0.10982 |
| P-Value | 0.80925 | 5.40E-10 | 0.1306 | 0.34466 | 3.08E-13 | 0.18882 | 0.84521 | 1.23E-08 | 0.16628 |
| $\alpha$ | 0.2 | 0.01 | 0.2 | 0.2 | 0.01 | 0.2 | 0.2 | 0.01 | 0.2 |
| Reject? | No | Yes | Yes | No | Yes | Yes | No | Yes | Yes |

**Main comment #5**

"Concerning the 7 scenarios reported in table 5, how have you defined them? Are there specific reasons behind the use of initial or optimal ranges for cases 5, 6 and 7? In addition, I suggest to keep the same column order for parameters, it's easier to read table 5 in relation to the values of table 4."

**Responses to main comment #5:**

Case 1 was defined as the initial case using all initial ranges. Cases 2-4 were defined as the single parameter range selection (S-SPR) cases. Cases 5-7 were defined as the multiple parameters ranges selections (M-SPR) cases.

The seven cases were set to demonstrate three primary results. Firstly, the M-SPR method is superior to the S-SPR one for calibrating hydrologic models with multiple parameters. It can be deduced from higher $E_{NS}$ values of Cases 5-7 than those of Cases 1-4. Secondly, merely using the optimal range of the parameter of relatively higher sensitivity contributes to more efficient calibration when the two parameters have negative effect on each other. It can be concluded by comparing the $E_{NS}$ values of

Cases 2-4 referring to the two parameters EX and Im. Thirdly, the combination of optimal ranges of all parameters is not the optimum inasmuch as some parameters like Im have negative effects on other parameters. It can be inferred through analyzing the $E_{NS}$ values of Cases 5-7. The analysis of sensitivity and correlation between parameters is, therefore, very important to determine the optimum ranges combination of all parameters for model calibration.

As for the column order of tables, we will modify Table 2, 5 so that the parameters are ordered in the same way as they do in Table 4.

**Main comment #6**

"The writing in some part of the manuscript should be improved. I suggest to carefully go through the overall manuscript and check verbs and syntax (here some example: P5, L3; P5, L23; P6, L5, P9, L25-26; ...; P10, L17)."

**Responses to main comment #6:**

We will revise the manuscript as the suggestion:

Page 5, line 3: "which representing agreement between observed and simulated data" >> "which represents the agreement between observed and simulated data"

Page 5, line 23: "…, the initial range of parameter is required adjusting properly" >> "…, the initial range of parameter requires adjusting properly"

Page 6, line 5: "... values can transform and finally convert into..." >> "... values can be converted into..."

Page 9, line25-26: "...when Case 4 compared with Case 1, Case 2 and Case 3 It can be explained..." >> "... when Case 4 is compared with Case 1, Case 2 and Case 3. It can be explained..."

Page 10, line 17: "to adopted" >> "to be adopted"

In addition, we will check the paper carefully and correct the other language errors. For example:

Page 1, line 3: "Qiaofeng." >> "Qiaofeng"

Page 1, line 14: "characteristics of single parameter value was analysed" >> "of single parameter value was analysed"

Page 1, line 17: "corresponding to the distribution" >> "corresponding to the distribution type"

Page 2, line 4: "mechanism of water cycle" >> "mechanism of the water cycle"

Page 2, line 9: "the streamflow at catchment outlet" >> "the streamflow at the catchment outlet"

Page 3, line 23-24: "single parameter is selected" >> "single parameter was selected"

Page 3, line 30: "in flood reason" >> "in flood season"

Page 4, line 18: "from observed streamflow" >> "from the observed streamflow"

Page 5, line 11: "whisker to the box" >> "the whisker to the box"

Page 5, line 28: "represents the ranges" >> "represent the ranges"

Page 6, line 4: "in case of larger percentage" >> "in case of a larger percentage"

Page 6, line 11: "may more and less effect" >> "may affect"

Page 6, line 13: "contributes" >> "contribute"

Page 6, line 15: "The index $R_c$ were quantified" >> "The index $R_c$ was quantified"

Page 6, line 17: "the greater positive influence" >> "greater positive influence"

Page 7, line 2: "The statistic analysis" >> "The statistical analysis"

Page 7, line 6: "ranges is substituted" >> "ranges are substituted"

Page 7, line 8: "the selected one is adopted for calibration of multiple parameters." >> "the selected ones are adopted for multi-parameters model calibration."

Page 7, line 16: "In stage 3 the ... " >> "In stage 3, the ..."

Page 7, line 26: "direction of Y axis" >> "direction of the Y axis"

Page 7, line 30-31: "The ratio of calibrated parameter range to initial one is less than 30% for parameters CI, SM, and Kc" >> "The ratios of the calibrated parameter range to the initial one are less than 30% of parameters CI, SM, and Kc"

Page 7, line 31: "It suggest that" >> "It suggests that"

Page 8, line 6: "To normal distribution" >> "For normal distribution"

Page 8, line 7: "different parameter range are selected" >> "different parameter ranges are selected"

Page 8, line 13: "concentrates at larger value zone" >> "concentrates at a higher value range"

Page 8, lien 24: "Because the parameter values in MINR indicate a high probability to be picked out to achieve high $E_{NS}$, vice versa." >> "It is because that the parameter values that may achieve a higher $E_{NS}$ can be easily picked out from the MINR of higher probability density."

Page 8, line 33: "box-plot chart of $E_{NS}$ for different ranges are shown in Fig. 8e" >> "box-plots for different ranges are shown in Fig. 8e"

Page 9, line 6: "value in columns" >> "values in columns".

Page 9, line 15: "… and CG are of high sensitive to $E_{NS}$" >> "... and CG are highly sensitive to $E_{NS}$"

Page 9, line 20: "penetrate" >> "penetrability"

Page 9, line 23: "there is contradiction owing to it" >> "there is a contradiction owing to it"

Page 9, line 25: "pf" >> "of"

Page 10, line 13: "the extension range followed by" >> "the extended range followed by"

**Specific comment #1**

"Abstract: in the last part of the abstract, roughly from line 20 on, the Authors report some specific methodological considerations that may not be really clear to one who has not already read the paper. I suggest to focus more on the scope and aims of the analysis, reporting also that the methodology proposes indexes for the evaluation of parameter sensitivity and correlations, as well as a summary of the main outcomes."

**Responses to specific comment #1:**

According to the comments of the referee, we will rewrite the abstract as follows:

The parameters are usually calibrated to achieve good performance of hydrological models, owing to the highly non-linear problem of hydrology process modelling. However, parameter calibration efficiency has a direct relation with parameter range. Furthermore, parameter range selection is affected by probability distribution of parameter values, parameter sensitivity and correlation. A newly proposed method is employed to determine the optimal combination of multi-parameter ranges for improving the calibration of hydrological models. At first, single-parameter probability distributions were analyzed based on 100 samples obtained from independent Genetic Algorithms (GA) calibration performed on a Xinganjiang model with a corresponding initial parameter range and, the distribution type (i.e. normal, exponential and uniform distributions) was specified for each parameter of the model. Then, the optimal range for each parameter was determined by comparing $E_{NS}$ values calculated

separately with the initial range, the minimum and maximum ranges of a given cumulative frequency of 50% (i.e. MINR and MAXR) and the extended range. Next, parameter correlation and sensibility were evaluated by quantifying two indexes $R_{C\,Y,X}$ and $S_E$ which can be used to coordinate with the negatively correlated parameters to specify the optimal combination of ranges of all parameters for

5    calibrating models. It is shown from the investigation that the probability distribution of calibrated values of any particular parameter in a Xinanjiang model is closely approximated by a normal or exponential distribution. The multi-parameter optimal range selection method is superior to the single-parameter one for calibrating hydrologic models with multiple parameters. The combination of optimal ranges of all parameters is not the optimum inasmuch as some parameters like Im have negative effects

10   on other parameters. The application of the proposed methodology gives a rise to an increase of 0.01 in minimum $E_{NS}$ compared with that of the pure GA method. The rising of minimum $E_{NS}$ with little change of the maximum may shrink the range of the possible solutions, which can effectively reduce uncertainty of the model performance.

15   **Specific comment #2**

"P1, L29: is "method" appropriate to indicate hydrological process modelling? I would suggest something like "tools" or similar."

**Responses to specific comment #2:**

We will replace "method" with "tool" in the first sentence of "Introduction".

**Specific comment #3**

"P4, L28: On which base you say that 100 samples are enough? Have you adopted some statistical texts to verify the statistical distribution of the considered parameters."

**Responses to specific comment #3:**

25   Before defining sampling size value, we performed a lot of trial tests. Figure D shows the variation curves of maximum and minimum $E_{NS}$ with sample size. It is indicated that both maximum and minimum $E_{NS}$ keep stable when sampling size is greater than 100. Avoiding the time-consuming computation, we assigned sampling size for the study as 100.

     With regard to the statistical distribution of the parameters, we performed the K-S tests to define the

30   distribution type for each parameter. The results of some K-S tests are given in the response to main comment #4.

[Figure]

Figure D. Variation curves of maximum and minimum $E_{NS}$ with sample size

**Specific comment #4**

"P4, L3: "A Genetic Algorithm (GA) was selected""

**Responses to specific comment #4:**

Actually, the sentence mentioned above appears in P4, L30. We will modify it as it is suggested.

**Specific comment #5**

"P9, L6: why do you say that it is obvious?"

**Responses to specific comment #5:**

According to the values in Table 4, $R$C value in columns of parameters CI and WM are positive, most $R$C values in column of parameter Im are negative. In order to make it easy to read, we will change it to "It is obvious from Table 4 that […]".

**Specific comment #6**

"P10, L7: please remove the colon;"

**Responses to specific comment #6:**

We will remove the colon in Line 7 of Page 10.

**Specific comment #7**

"Fig. 2: check "curve"; I also suggest to re-word the caption as: [...]; Cumulative frequency and [...] distribution for normal (b), exponential (c) and uniform (d) distributions."

**Responses to specific comment #7:**

We will replace "cure" with "curve" in the caption of Figure 2. In addition, we will modify the caption of Figure 2 as suggested: "…; Cumulative frequency curve and histogram for normal (b), exponential (c) and uniform (d) distributions".

**Specific comment #8**

"Fig. 6, 7 and 8: is it necessary to report the label "schema"?"

**Responses to specific comment #8:**

5 We will remove the label "schema" in Figs. 6a, 6b, 6c, 7a, 7b, 7c, 8a, 8b, 8c and 8d.

**Specific comment #9**

"Table 1: is P the average or the max?"

**Responses to specific comment #9:**

10 We will modify the note on Table 1 as follows: "$Q_{Max}$, $Q_{Min}$ and $Q_{Avg}$ mean the maximum, minimum and average value of daily streamflow, respectively, and $P_{Max}$ means the maximum value of daily precipitation.". Meanwhile, we will modify the corresponding description in section 2 to avoid the misunderstanding.

15 **Specific comment #10**

"Table 2: the definition of parameter B seems not complete. Also, the column "range" of Table 2 is reported twice (see Table 3)."

**Responses to specific comment #10:**

We will complete the definition of parameter B in the modified Table 2 presented below. The column
20 "range" of Table 2 will be changed as column "units" because the ranges of parameters are reported in Table 3.

Table 2. Parameters of Xinanjiang model

| Parameter | Definition | Units |
|---|---|---|
| CI | Recession constants of the lower interflow storage | dimensionless |
| Kc | Ratio of potential evapotranspiration to pan evaporation | dimensionless |
| KI | Outflow coefficients of the free water storage to interflow | dimensionless |
| SM | Areal mean free water capacity of the surface soil layer, which represents the maximum possible deficit of free water storage | mm |
| B | Exponential parameter with a single parabolic curve, which represents the non-uniformity of the spatial | dimensionless |
| WM | Averaged soil moisture storage capacity of the whole layer | mm |
| C | Coefficient of the deep layer, that depends on the proportion of the basin area covered by vegetation with deep roots | dimensionless |
| EX | Exponent of the free water capacity curve influencing the development of the saturated area | dimensionless |
| CG | Recession constants of the groundwater storage relationships | dimensionless |
| KG* | Outflow coefficients of the free water storage to groundwater relationships | dimensionless |
| Im | Percentage of impervious and saturated areas in the catchment | dimensionless |

* the value of KG is calculated by the function 0.7-KI

**Specific comment #11**

"Table 3: the main legend is not really clear, I suggest to re-word it. ** "ratio of calibrated parameter ...""

5    **Responses to specific comment #11:**

The definition of "Ratio" in Table 3 will be modified as follows: "** the ratio is calculated by dividing the parameter range derived from 100 GA calibration by the initial parameter range".

**Part 2 Relevant Changes Following Referees' Comments**

1. Following "Referee #1 Main comment #1", "Referee #2 Main comment #4" and "Referee #2 Specific comment #3" comments, we have modified sections 3.2.2 & 4.1 and Table 3. The following were added and/or modified:

   a. Page 5, line 8-20:

   The probability distributions of calibrated parameters values can be estimated roughly by using box-plot charts, cumulative frequency curves and frequency histograms. The symmetry of the box-plot chart (including one box and two whiskers) and the length ratio of the whisker to the box, the shape of the cumulative frequency curve and the frequency histogram are important indicators for the identification of the distribution type. Based on these indicators, three types of probability distributions are listed as follows: (1) Normal distributions, the box and whiskers are approximately symmetrical along the Y-axis direction, the length of either whisker is longer than half height of the box in a box-plot chart (Fig. 3a), the cumulative frequency curve is S shaped and the histogram bell shaped (Fig. 3b); (2) Exponential distributions, the whole chart is distinctly asymmetrical in the Y-axis direction which means that the average value (marked with a small hollow square) deviates from the median value (marked with a centre line in box), the box is inclined to one side with the extreme shorter whisker (Fig. 3a), the cumulative frequency curve is parabola shaped, and the histogram tends to increase or decline gradually (Fig. 3c); (3) Uniform distribution, the box and whiskers are approximately symmetrical along the Y-axis direction, the length of two whiskers is close to that of the box (Fig. 3a), the cumulative frequency curve tends to a straight line and the histogram varies little along the X-axis (Fig. 3d).

   b. Page 5, line 21:

   "A Kolmogorov-Simirnov test (K-S test) tries to examine whether a data set fit a reference probability distribution or not (Haktanir, 1991). In a K-S test, for any variable $x_i$ in a data set, the empirical distribution function value ($Fi$) is calculated by using a plotting position formula, and the cumulative distribution function value ($Fi^*$) is computed by using the reference probability distribution. The maximum deviation between the two values, $\Delta_{Max}$, is expressed in Eq. (2).

   $$\Delta_{Max} = |F_i^* - F_i| \tag{2}$$

   According to the acceptable level of significance $\alpha$ ($\alpha$=0.2) and the total number of values in a data set n, $\Delta_{table}$ can be obtained from the K-S table. If $\Delta_{Max} < \Delta_{table}$, the reference probability distribution is identified to fit to the data set."

   c. Page 7, line 25-27:

   "It is obvious that the box and whiskers are symmetrical and the length of whiskers is longer than that of box along the direction of Y axis for parameter CI, SM and Kc." >> " It is obvious from Fig. 7 that the box and whiskers are approximately symmetrical and the length of whiskers is longer than that of half box along the direction of the Y axis for parameters CI, SM and Kc."

   d. Page 7, line 30:

   "Furthermore, K-S tests were employed to determine the probability distributions of

parameters and the corresponding results are listed in Table 3. It is shown that only a normal distribution is accepted for parameters CI & SM. Despite the fact that both normal and uniform distributions are accepted for parameter KC, the probability distribution of parameter KC is regarded as a normal distribution. It is because that the $\Delta_{Max}$ will become smaller if a normal distribution serves as a reference distribution instead of a uniform distribution. In addition, just an exponential distribution is accepted for the rest of the parameters. Thus, the three parameters follow normal distributions and the others exponential distributions in the Xinanjiang model."

e. Page 23:

Table 3. Range changes and K-S tests ($\alpha$=0.2) of parameters in schema Initial

| Parameter | Initial range | Calibrated range* | Ratio** (%) | $\Delta_{Max}$*** | | |
| --- | --- | --- | --- | --- | --- | --- |
| | | | | Normal distribution | Expoential distribution | Uniform distribution |
| CI | 0–0.9 | 0.630–0.745 | 12.78 | 0.062 (pass) | 0.328 (fail) | 0.115 (fail) |
| Kc | 0–1.1 | 0.81–1.09 | 25.45 | 0.076 (pass) | 0.305 (fail) | 0.089 (pass) |
| KI | 0–0.7 | 0.534–0.7 | 23.71 | 0.128 (fail) | 0.076 (pass) | 0.173 (fail) |
| SM | 10–50 | 31–39.4 | 21.00 | 0.060 (pass) | 0.304 (fail) | 0.110 (fail) |
| B | 0.1–0.4 | 0.238–0.4 | 54.00 | 0.180 (fail) | 0.062 (pass) | 0.203 (fail) |
| WM | 120–200 | 120–150 | 37.50 | 0.181 (fail) | 0.072 (pass) | 0.231 (fail) |
| C | 0.1–0.2 | 0.1–0.2 | 100.00 | 0.163 (fail) | 0.082 (pass) | 0.217 (fail) |
| EX | 1.0–1.5 | 1.0–1.5 | 100.00 | 0.118 (fail) | 0.079 (pass) | 0.135 (fail) |
| CG | 0.950–0.998 | 0.950–0.994 | 91.67 | 0.123 (fail) | 0.102 (pass) | 0.139 (fail) |
| Im | 0.01–0.04 | 0.01–0.04 | 100.00 | 0.134 (fail) | 0.076 (pass) | 0.148 (fail) |

* the calibrated parameter range except the extreme outlier

** the ratio is calculated by dividing the length of the range derived from 100 GA calibration runs by the initial range length

*** the $\Delta_{Max}$ is calculated by using the normalnized parameter values

2. Following "Referee #1 Main comment #3" and "Referee #2 Main comment #6" comments, we have revise the manuscript as the suggestion. In addition, we have checked the paper carefully and corrected the other language errors. The following were added and/or modified:

Page 1, line 3: "Qiaofeng." >> "Qiaofeng"

Page 1, line 14: "characteristics of single parameter value was analysed" >> "of single parameter value was analysed"

Page 1, line 17: "corresponding to the distribution" >> "corresponding to the distribution type"

Page 2, line 2: "The hydrological model is a type of black-box model in 1932 originally (Sherman, 1932), and conceptual models and distributed models are subsequently put forward in 1960s (Freeze and Harlan, 1969)." >> "The initial hydrological model was a black-box model in 1932 (Sherman, 1932) and conceptual & distributed models are subsequently put forward in 1960s (Freeze and Harlan, 1969)."

Page 2, line 4: "mechanism of water cycle" >> "mechanism of the water cycle"

Page 2, line 6: "the interflow and base flow are simplified" >> "the interflow and the base flow are simplified"

Page 2, line 9: "the streamflow at catchment outlet" >> "the streamflow at the catchment outlet"

Page 2, line 15-16: "obtain exact optimal solution" >> "obtain an exact optimal solution"

Page 2, line 17: "mathematical methods, having wide application in …" >> "mathematical calculations, having a wide application in …"

Page 2, line 25: "having powerful capability" >> "having a powerful capability"

Page 2, line 27: "search for the optimal solution" >> "search for the an optimal solution"

Page 2, line 30: "of the hydrological model" >> "of a hydrological model"

Page 2, line 31-32: "In general, parameter variables obey some types of probability distribution in the given range after multiple independent repeat calibration by an auto-calibration method" >> "In general, parameter variables obey some special probability distributions within the given range after multiple independent calibration"

Page 2, line 32-33: "Graziani et al. (2008) stated that the shape of the parameter value probability distributions can be significantly affected by their ranges. " >> "Graziani et al. (2008) stated that the shape of a parameter probability distribution can be significantly affected by a parameter range. "

Page 3, line 2: "Although Normal …" >> "Although normal …"

Page 3, line 7-8: "calibration of other parameters correlated with it" >> "calibration of other related parameters"

Page 3, line 9: "varies with catchment characteristic, objective function …" >> "varies with catchment characteristics, objective functions …"

Page 3, line 10: "parameter ranges lead to" >> "parameter ranges could lead to"

Page 3, line 11: "reducing or extending the ranges would affect the parameters sensitivity, making insensitive parameters …" >> "reducing or extending ranges might make insensitive parameters …"

Page 3, line 15-16: "The more deviation between true ranges and given range, the more instability of calculated result." >> "The more deviation between an optimal range and a given range, the more uncertainty of the calculation results."

Page 3, line 16: "Appropriate parameter ranges selection is ..." >> "The selection of appropriate parameter ranges is …"

Page 3, line 16: "few literature reported" >> "few literature covers information on"

Page 3, line 21-23: "At first, probability distribution characteristics of parameter values were analysed based on the parameter value samples that calibrated by using a GA method." >> "At first, probability distribution of each parameter was analysed based on a lot of independent calibrations by using a GA method."

Page 3, line 23-24: "range of single parameter is selected" >> "range of a single parameter was specified"

Page 3, line 30: "in flood reason" >> "in flood season"

Page 3, line 31: "The thickness of soil varies in most karst areas tremendously different with space: limestone exposed in some peak-cluster region, 2-10 m thickness clay covered in the depression and valley bottom." >> "The thickness of soil varies spatially in most karst areas. Limestone is exposed to air in some peak-cluster region. Clay soil with thickness ranging from 2 to 10m is distributed in the depressions and valleies."

Page 4, line 8-10: "The maximum areal precipitation of the studied catchment varies with year, the value is 235 mm/d of 1996 while 107 mm/d of 2000. The average streamflow decreases from 14.38 to 11.37 m3/d during the studied period." >> "The maximum areal daily precipitation varied with years in the studied catchment and reached the value of 235 mm/d in 1996."

Page 4, line 13: "parameters ranges selections" >> "parameters ranges selection"

Page 4, line 18: "from observed streamflow" >> "from the observed streamflow"

Page 4, line 19: "the meaning and the common range of …" >> "the definitions of …"

Page 4, line 20-21: "The proposed PRS method is introduced as follows, taking a Xinanjiang model for example." >> "The proposed PRS method is introduced as follows, when a Xinanjiang model is taken as an example."

Page 4, line 24: "In theory, the results of calibration by using a stochastic-based auto-calibration method are not completely same but similar in a reasonable convergence condition, which obey some probability distributions." >> "In theory, the  parameter values calibrated by using a stochastic-based auto-calibration method are not same to each other but obey a certain probability distribution under a reasonable convergence condition."

Page 4, line 28: "… calibrated parameter values" >> "… calibrated parameter values in the investigation"

Page 4, line 30: "because GAs are common …" >> "because GA is a common …"

Page 4, line 31-32: "Many studies showed that the evolutionary algorithms could provide equal or better performance than other algorithms" >> "Many studies show that evolutionary algorithms provide equal or better performance of a model than other algorithms do"

Page 5, line 3: "which representing agreement between observed and simulated data" >> "which represents the agreement between observed and simulated data"

Page 5, line 5-6: "… observed streamflow … observed streamflow … simulated streamflow … mean value …" >> "… the observed streamflow … the observed streamflow … the simulated streamflow … the mean value …"

Page 5, line 11: "whisker to the box" >> "the whisker to the box"

Page 5, line 23: "…, the initial range of parameter is required adjusting properly" >> "…, the initial range of a parameter requires adjusting properly"

Page 5, line 24: "the different ways to adjust specify the optimal ranges for a single parameters" >> "the different ways to adjust specify the optimal ranges for a single parameter"

Page 5, line 25: "For uniform distribution, it is better to keep initial range due to little influence of the range on calibration results." >> "For the parameter of a uniform distribution, it is better to keep the initial range due to little influence of ranges on calibration results."

Page 5, line 25-26: "For normal distribution, the cumulative frequency curve is employed to seek several of reduced ranges …" >> "For the parameter of a normal distribution, the cumulative frequency curve is employed to seek some reduced ranges …"

Page 5, line 28: "represents the ranges …with a given cumulative frequency …" >> "represent the ranges …under a given cumulative frequency …"

Page 5, line 29-30: "As for exponential distribution, the initial range can be doubled from the boundary of high probability density to the outside, if the parameter has reasonable meaning in the new range." >> "As for the parameter of an exponential distribution, the initial range is

symmetrically duplicated on one side of high probability density, if the parameter has reasonable meaning in the extended range."

Page 5, line 30-31: "Thus, the exponential distribution can be converted into normal distribution and then the optimal range can be selected by using the method for normal distribution." >> " Then, the optimal range of the parameter can be specified by comparing different ENS calculated separately by using the initial range, the MINR or MAXR of the initial range, the MINR or MAXR of the extended range."

Page 6, line 5: "... values can transform and finally convert into..." >> "... values can be converted into..."

Page 6, line 8-10: "As far as the Xinanjiang model … both parameter WM and B refer to the water storage volume – area curve that representing ..." >> "As far as a Xinanjiang model … parameters WM and B refer to the water storage volume – area curve that represents ..."

Page 6, line 10: "If the curve is fixed, a larger WM results in a smaller B" >> "If the curve is fixed, the larger WM results in the smaller B"

Page 6, line 10-11: "The change of a parameter range may more or less effect the calibration of other parameters." >> "As a result, the range change of parameter WM may affect the range setting and calibration of parameter B."

Page 6, line 11: "if several parameters ranges require adjusting" >> "if the ranges of the related parameters require adjusting"

Page 6, line 13: "If the change of one parameter … other parameters, the selected ranges for the parameter will contributes to ..." >> " If the range change of one parameter … other parameters, using the optimal range of the parameter instead of the initial one can contribute to ..."

Page 6, line 14: "… the negative influence may make the contribution of the selected ranges against model calibration." >> "… the negative impact may result in a worse model calibration, although the optimal ranges of the parameters are used."

Page 6, line 15-16: "The index $Rc$ were quantified quantified to analyse the influencing degree of one parameter range …" >> "The index $Rc$ was quantified to analyse the influence degree of one-parameter range …"

Page 6, line 16-18: "The more close value of $R_{C\,Y,X}$ to 1, the greater positive influence of range change of parameter X on calibration of parameter Y. If $R_{C\,Y,X}$ less than 0 ..." >> "When $R_{C\,Y,X}$ is closer to 1, the range change of parameter X has a greater positive influence on the calibration of parameter Y. If $R_{C\,Y,X}$ is minus ..."

Page 6, line 20: "the influencing degree of range change … on calibration of parameter Y" >> "the influence degree of the range … on the calibration of parameter Y"

Page 6, line 22-23: "selected range" >> "the optimal range"

Page 6, 24-26: "If there is negative influence between two parameters, the parameter of high sensitivity is ranked as primary one and its selected ranges can be kept in the range combination for all parameters, while the initial range is used in place of the selected range to minimize the negative effect for the other parameter of low sensitivity." >> "If there is a negative influence between two parameters, the optimal range of the parameter of higher sensitivity is used and the initial range of the other parameter kept for calibration generally to mitigate the negative impact."

Page 6, line 26: "It is due to the fact that" >> "It is due to that"

Page 6, line 27: "… parameters do during multi-parameter calibration …" >> "… parameters do in a multi-parameter calibration …"

Page 6, line 28: "by performing S-PRS method" >> "by performing a S-PRS method"

Page 6, line 29-30: "The larger $R_E$, the more concentrated $E_{NS}$ distribution, which means parameter calibration is stable and efficient. Thus, the parameter of high $S_E$ is given priority to use the selected range ..." >> "The larger value of $R_E$, the more concentrated distribution of $E_{NS}$ which means more efficient parameter calibration. Thus, the parameter of higher $S_E$ is given priority to use the optimal range ..."

Page 7, line 2: "with initial range … with selected range … The statistic analysis …" >> "with an initial range … with an optimal range … The statistical analysis…"

Page 7, line 5: "range selection are investigated" >> "range selection were investigated"

Page 7, line 7: "of one parameter range change" >> "of one-parameter range change"

Page 7, line 6: "ranges is substituted" >> "ranges are substituted"

Page 7, line 8: "the selected one is adopted for calibration of multiple parameters." >> "the optimal ones are adopted for the multi-parameters calibration."

Page 7, line 16: "In stage 3 the ... " >> "In stage 3, the ..."

Page 7, line 20: "of Xinanjiang model" >> "of the Xinanjiang model"

Page 7, line 21: "100 times independent calibration" >> "100 independent calibration runs"

Page 7, line 24-25: "The 100 calibrated values for single parameters were normalized by dividing them by the corresponding initial range, and the box-plot chart of the results is shown in Fig. 5. It is obvious 7 that … " >> "For any particular parameter, calibrated values were normalized by dividing a deviation between a calibrated value and the lower limit of the initial range by the length of the initial range. Based on 100 calibrated values after normalization, a box plot for a parameter is depicted. It is obvious from Fig. 7 that …"

Page 7, line 26: "direction of Y axis" >> "direction of the Y axis"

Page 7, line 30-31: "The ratios of calibrated parameter range to the initial one are less than 30% of parameters CI, SM, and Kc, while the ratio varies from 23% to 100% for parameters such as KI, B, CG, and Im." >> "The ratio of the calibrated range length to the initial range length is less than 30% for parameters CI, SM, and Kc, while the ratio exceeds 30 % for parameters B, WM, C, EX, CG, and Im."

Page 7, line 31: "It suggest that" >> "It implies that"

Page 8: "MINR >> the MINR" & "MAXR" >> "the MAXR" & "in case of" >> "in the case of"

Page 8, line 1-2: "that reducing the ranges is suitable to improve calibration for parameters whose values obey normal distributions, whereas that is not enough for parameters whose values obey exponential distributions." >> "that reducing the initial ranges can improve the calibration for parameters whose values obey normal distributions."

Page 8, line 6: "To normal distribution, reducing the range is generally used to select the appropriate range." >> "For a normal distribution, reducing the range was used to find the optimal range."

Page 8, line 7: "different parameter range are selected" >> "different ranges are selected"

Page 8, line 10-11: "whereas the normal distribution is changed to the exponential one when the range is cut to MAXR." >> "whereas the probability distribution approximates an exponential one when the MAXR is used."

Page 8, line 13: "concentrates at larger value zone" >> "concentrates at a higher value range"

Page 8, line 13-14: "It is indicated that the reduced range of high probability density is helpful to make calibration more steady and efficient." >> "Using the reduced range of high probability density is, therefore, helpful to make calibration more stable and more efficient."

Page 8, line 16-18: "Figure 7 shows the calibration results of parameter KI. Since the initial range of parameter KI can not be extended, the reduced range was searched by using the cumulative frequency curve, the MINR (0.660–0.700) and MAXR (0.522–0.660) were picked out." >> "Figure 9 shows the calibration results under three different input ranges of parameter KI. Since the initial range of parameter KI cannot be extended, the two reduced ranges (i.e. the MINR (0.660 – 0.700) and the MAXR (0.522 – 0.660)) were picked out by using the cumulative frequency curve."

Page 8, line 19-20: "… parameter KI values is converted from exponential distribution to uniform distribution when the initial range is reduced to MINR, whereas the exponential distribution is still kept when the range is cut to MAXR." >> "…parameter KI values is similar to a uniform distribution in the case of the MINR, whereas that is still exponential in the case of the MAXR."

Page 8, line 20-22: "The contribution of parameter ranges to $E_{NS}$ is shown in Fig. 7d. Similar to the results of parameter CI, MINR is best for calibration when compared with MAXR or initial range." >> "The contributions of the three parameter ranges to $E_{NS}$ are shown in Fig. 9d. Thus, the MINR is best for calibration of parameter KI when compared with the MAXR or the initial range, which is similar to the calibration result of parameter CI."

Page 8, lien 24: "Because the parameter values in MINR indicate a high probability to be picked out to achieve high $E_{NS}$, vice versa." >> "It is because that the parameter values that may achieve a higher $E_{NS}$ can be easily picked out from the MINR of higher probability density."

Page 8, line 31-32: "… the probability distribution of parameter B values is converted into approximate uniform distribution when the range is reduced from initial range to MINR or from the extended range to extension-MINR." >> "… the probability distribution of parameter B approximates a uniform distribution when the MINR or the extension-MINR is used."

Page 8, line 33: "box-plot chart of $E_{NS}$ for different ranges are shown in Fig. 8e" >> "box-plots for different ranges are shown in Fig. 10e"

Page 9, line 5: "The S-PRS method was employed to select the one-parameter optimal range for each parameter, and the optimal ranges, indexed $R_C$ and $S_E$ values are listed in Table 4." >> "The S-PRS method was employed to determine the optimal range for each parameter. According to the optimal ranges and the corresponding initial ranges, indexed $R_C$ and $S_E$ were quantified to understand parameter correlation and sensitivity."

Page 9, line 6: "value in columns of parameter CI and WM" >> "values in the columns of parameters CI and WM".

Page 9, line 10-12: "Parameter CI has the maximum RC $_{mean}$ of 0.465, while parameter Im the minimum RC $_{mean}$ of –0.026. Furthermore, RC mean values for all parameters are positive except for that for parameter Im. It is due to the accumulative negative influence of parameter Im on others." >> "Parameter CI has the maximum $R_{C\ mean}$ of 0.465 and parameter Im the minimum $R_{C\ mean}$ of –0.026. Furthermore, all parameters have positive $R_{C\ mean}$ values except for parameter Im, owing to the accumulative negative correlation between parameter Im and the others."

Page 9, line 13: "To coordinate the contradiction between parameters, the index $S_E$ is used to pick parameters of high sensitivity to $E_{NS}$." >> "To coordinate with negatively related parameters, the index $S_E$ was used to pick out parameters of higher sensitivity to $E_{NS}$."

Page 9, line 15: "It suggests that parameters CI, B, SM, KI, $K_c$, WM and CG are highly sensitive to $E_{NS}$, and parameters C, EX and Im of low sensitivity for $E_{NS}$. CI is the most sensitive parameter while Im the most insensitive parameter …" >> "It suggests that parameters C, EX and Im are of low sensitivity to $E_{NS}$ and the others highly sensitive to $E_{NS}$. Parameter CI is the most sensitive while Im the most insensitive …"

Page 9, line 20: "penetrate" >> "penetrability"

Page 9, line 22-23: "It can be deduced that the optimal range of insensitive parameter Im cannot be taken into account when there is a contradiction owing to it, in order to improve calibration." >> "Thus, the optimal ranges of parameters of higher sensitivity should be used to improve calibration."

Page 9, line 24: "seven cases are investigated" >> "seven cases were investigated"

Page 9, line 25: "pf" >> "of"

Page 9, line 25: "The results of seven cases are compared in Fig. 9." >> "The box plots of $E_{NS}$ for different cases are given in Fig. 11."

Page 9, line25-26: "...when Case 4 compared with Case 1, Case 2 and Case 3 It can be explained..." >> "... when Case 4 is separately compared with Case 1, Case 2 and Case 3. It can be explained..."

Page 9, line 27: "As $S_E$ of parameter Im is less than that of parameter EX, parameter EX is given priority to select the optimal range," >> "As the $S_E$ value of parameter Im is less than that of parameter EX, parameter EX is given priority to use the optimal range."

Page 9, line 31-page 10, line 2: "The box and whisker of $E_{NS}$ for Case 6 rise, which means Case 6 has a better performance of calibration than Case 5 does, when the optimal range of parameter CG is included. But the box and whisker of $E_{NS}$ for Case 7 decline when the optimal range of parameter Im is included. Because the mean $R_C$ value of parameter Im is negative and its $S_E$ much less than that of others, using the optimal range of Im is adverse to multi-parameter combined calibration." >> "Case 6 has the most concentrated values of $E_{NS}$ and the largest mean value of $E_{NS}$ among the three cases. It means that the combination of optimal ranges of all parameters (see Case 7) is not the optimum to calibrate a multi-parameter model inasmuch as some parameters like Im have negative correlation on other parameters. Hence, the initial ranges of parameters having negative mean values of $R_C$ and low values of $S_E$ are supposed to be used to calibrate parameters instead of the corresponding optimal ranges."

Page 10, line 4-7: "Considering that there is the relation between the parameter ranges and probability distributions of parameter value, an approach to determine the optimal range combination for multi parameters of hydrological models is put forward by analysing the parameter value probability distribution, parameter sensitivity and parameter correlation. A case of improving the calibration of the GA-based Xinanjiang model for karst areas is studied, and some findings are presented as follows." >> "Considering that there is a relation between the selection of multi-parameter ranges and the calibration effect of a hydrological model, an approach to determine an optimal combination of ranges for the multi-parameter calibration was

put forward by analysing parameter probability distribution, parameter sensitivity and correlation between parameters. The newly proposed method was applied for the calibration of a Xinanjiang model for karst areas, and some findings are presented as follows."

Page 10, line 13: "the extension range followed by MINR is recommended" >> "the extension-MINR is recommended"

Page 10, line 17: "to adopted" >> "to be adopted"

Page 10, line 18-19: "The investigation is financially supported by special funds for scientific research on public causes of Chinese Ministry of Water Resources" >> "The investigation is supported by special funds for public welfare industry research projects of the Ministry of Water Resources of the People's Republic of China"

**3.** Following "Referee #1 Specific comment #1" and "Referee #2 Specific comment #3" comments, we have expanded sections 3.2.1 & 3.3.1 and add Figures 2 & 5 to clarify more on how to determine the sampling size and cumulative frequency value. The following were added and/or modified:

a. Page 4, line 27-29:

"As far as the sample size is concerned, 100 samples are enough to estimate the probability distribution of calibrated parameter values in the investigation, which is deduced from the results of trial tests as shown in Fig. 2. It can be seen that both maximum and minimum $E_{NS}$ keep stable when sampling size is greater than 100."

b. Page 15:

[Figure]

Fig. 2. Variation curves of maximum and minimum $E_{NS}$ with sample sizes

c. Page 6, line 4:

"Figure 5 gives the variation curves of maximum and minimum $E_{NS}$ of a single parameter with cumulative frequency values. It is found that the maximum $E_{NS}$ keeps constant despite a cumulative frequency value varying, while the minimum $E_{NS}$ approaches the peak value of 0.881 when the cumulative frequency value is equal to 50%. Considering that higher minimum $E_{NS}$ contributes to more efficient calibration, the fixed cumulative frequency value of 50% was selected to determine the ranges of maximum and minimum probability density (i.e. MINR and MAXR) for each parameter."

d.  Page 16:

[Figure]

Fig. 5. Variation curves of maximum and minimum $E_{NS}$ of a single parameter with cumulative frequency values

5   **4.** Following "Referee #1 Specific comment #2" comments, we have revise the result section as the
suggestion. The following were added and/or modified:
a.  Page 8, line 12:
"It is found that the minimum $E_{NS}$ except extreme outliers rises convincingly and $E_{NS}$
concentrates at larger value zone when MINR is used instead of the initial range." >> " It is
10            found that the minimum $E_{NS}$ except extreme outliers rises from 0.881 to 0.884 and the $E_{NS}$
concentrates at a higher value range when the MINR is used instead of the initial range."
b.  Page 8, line 34:
"It is shown that there is little improvement in maximum $E_{NS}$ when MINR is used for
calibration instead of the initial range. There is an increase of 0.0003 in maximum $E_{NS}$ if the
15            initial range is replaced with the extension range or the extension-MINR. As for minimum
$E_{NS}$ (except outliers), an increase of 0.001 in the case of the MINR, a decrease of 0.003 in
the case of the extension range and an increase of 0.003 in the case of the extension-MINR
are found when the initial range is substituted with the three ranges respectively."
c.  Page 9, line 29-30:
20            "As for the cases of multi-parameter range selection (i.e. Case 5, Case 6 and Case 7), the
results are much better than of Case 1-4."   >> "As for the cases with the multi-parameter
range selection (i.e. Cases 5－7), the $E_{NS}$ values are much greater than those of cases 1-4.
There is approximately an increase of 0.001 in maximum $E_{NS}$ and an increase of 0.01 in
minimum $E_{NS}$ when the multi-parameter range selection is performed."

**5.** Following "Referee #1 Specific comment #3" and "Referee #2 Main comment #5" comments, we
have modified section 4.3 & 5. The following were added and/or modified:
Page 9, line 25:
"Case 1 was defined as the initial case using all initial ranges. Cases 2－4 were defined as the
30            single parameter range selection (S-SPR) cases. Cases 5－7 were set as the multiple parameters
ranges selections (M-SPR) cases."
Page 10, line 8-17:

"In the Xinanjiang model for karst areas, parameters CI, Kc, SM and B approximately obey normal probability distributions, and parameters WM, C, EX, KI, CG and Im exponential probability distributions after 100 independent calibration runs. For the parameters of a normal distribution, the MINR defined by using a cumulative frequency curve of calibrated values is preferred to be selected as the optimal parameter range for calibration. For the parameters of an exponential distribution, the extension-MINR is recommended to be used for calibration if the initial range can be extended towards the high-probability side, otherwise the MINR is selected as the optimal range for calibration.

The proposed parameter range selection (PRS) method improves the minimum and mean values of $E_{NS}$. The application of the proposed methodology results in an increase of 0.01 in minimum $E_{NS}$, compared with that of the pure GA method. The rising of minimum $E_{NS}$ with little change of the maximum may shrink the range of the possible solutions. As a result, the uncertainty of the model performance can be effectively controlled.

The M-SPR method is superior to the S-SPR one for calibrating hydrologic models with multiple parameters. The $R_C$ and $S_E$ are two important indexes that can help to analyse the sensitivity and correlation between parameters and consequently to coordinate with the negatively related parameters. The initial ranges of parameters of relatively low $S_E$ and negative $R_{C\ mean}$ and the optimal ranges of parameters of positive $R_{C\ mean}$ should be preferred to be chosen for the multi-parameter model calibration."

6. Following "Referee #1 Specific comment #4" comments, we have revise Figure 1 as the suggestion. The following were added and/or modified:
Page 14:

[Figure]

Fig. 1. Location of the study area

7. Following "Referee #1 Specific comment #5" and "Referee #2 Specific comment #7" comments,

we have revise the caption of Figure 2 as the suggestion. The following were added and/or modified:
Page 15, line 13:

"(a) Box-plot charts of normal, exponential and uniform distribution; Cumulative frequency curve and histogram for normal (b), exponential (c) and uniform (d) distributions"

**8.** Following "Referee #1 Specific comment #6" comments, we have revise Figure 5 as the suggestion. The following were added and/or modified:
Page 17:

[Figure]

10               Fig. 7. The box-plot chart of normalized calibrated values for parameters of Xinanjiang model

**9.** Following "Referee #1 Specific comment #7", "Referee #1 Specific comment #8" and "Referee #2 Specific comment #10" comments, we have modified Table2. The following were added and/or modified:

15       Page 22:

Table 2. Parameters of Xinanjiang model

| Parameter | Definition | Units |
|:---:|:---|:---:|
| CI | Recession constants of the lower interflow storage | dimensionless |
| Kc | Ratio of potential evapotranspiration to pan evaporation | dimensionless |
| KI | Outflow coefficients of the free water storage to interflow | dimensionless |
| SM | Areal mean free water capacity of the surface soil layer, which represents the maximum possible deficit of free water storage | mm |
| B | Exponential parameter with a single parabolic curve, which represents the non-uniformity of the spatial | dimensionless |
| WM | Averaged soil moisture storage capacity of the whole layer | mm |
| C | Coefficient of the deep layer, that depends on the proportion of the basin area covered by vegetation with deep roots | dimensionless |
| EX | Exponent of the free water capacity curve influencing the development of the saturated area | dimensionless |

| | | |
|---|---|---|
| CG | Recession constants of the groundwater storage relationships | dimensionless |
| KG* | Outflow coefficients of the free water storage to groundwater relationships | dimensionless |
| Im | Percentage of impervious and saturated areas in the catchment | dimensionless |

\* the value of KG is calculated by the function 0.7-KI

**10.** Following "Referee #2 Main comment #5" comments, we have revise Table 5 as the suggestion. The following were added and/or modified:

Page 24:

Table 5. Parameter ranges setting for different cases

| Case | Range setting of parameter | | | | | | | | | |
|---|---|---|---|---|---|---|---|---|---|---|
| | CI | Kc | KI | SM | B | WM | C | EX | CG | Im |
| 1 | I | I | I | I | I | I | I | I | I | I |
| 2 | I | I | I | I | I | I | I | I | I | O |
| 3 | I | I | I | I | I | I | I | O | I | I |
| 4 | I | I | I | I | I | I | I | O | I | O |
| 5 | O | O | O | O | O | O | O | O | I | I |
| 6 | O | O | O | O | O | O | O | O | O | I |
| 7 | O | O | O | O | O | O | O | O | O | O |

The symbol 'I' represents the initial range of the parameter in Table 3, and 'O' the optimal range of the parameter in Table 4.

**11.** Following "Referee #2 Specific comment #11" comments, we have revise the main legend of Table 3 as the suggestion. The following were added and/or modified:

Page 21, line 3:

"** the ratio is calculated by dividing the parameter range size derived from 100 GA calibration by the initial parameter range size"

**12.** Following "Referee #2 Main comment #2" comments, we have expanded sections 3.2.1 & 4.3 and add Figure 11 to clarify more on GA parameters and computational/time efforts. The following were added and/or modified:

a.  Page 7, line 21:

"Trial tests were employed to determine the optimal GA control parameters: crossover probability of 0.5, mutation probability of 0.7 for the individual, mutation probability of 0.5 for each gene, population size of 21, maximum generation number of 500 and maximum iteration number of 50. These parameters were kept constant for GA calibrations in the investigation."

b.  Page 10, line 3:

"Through a calibration run, a set of calibrated values of all parameters and the corresponding $E_{NS}$ are obtained. Figure 12 shows the variation curves of maximum and minimum values of $E_{NS}$ with number of runs by using a GA method and a proposed PRS method, respectively. It is indicated from Figure 12 that no mater it is maximum or minimum $E_{NS}$, the value calculated by using a proposed method is almost the same as that by using a GA method when the number of runs does not exceed 100. If a proposed method is used for calibration

instead of a GA method, there are approximately an increase of 0.001 in maximum $E_{NS}$ and an increase of 0.01 in minimum $E_{NS}$ when the number of runs is greater than 100. Thus, for any particular run number, the value of $E_{NS}$ calculated by using a PRS method is not less than that by using a GA method. The application of a proposed method, therefore, contributes to a relatively efficient calibration."

c. Page 21:

[Figure]

Figure 12. The variation curves of maximum and minimum $E_{NS}$ with number of runs by using a GA method and a proposed PRS method

**13.** Following "Referee #2 Specific comment #1" comments, we have revise the abstract as the suggestion. The following were added and/or modified:

Page 1:

"The parameters are usually calibrated to achieve good performance of hydrological models, owing to the highly non-linear problem of hydrology process modelling. However, parameter calibration efficiency has a direct relation with parameter range. Furthermore, parameter range selection is affected by probability distribution of parameter values, parameter sensitivity and correlation. A newly proposed method is employed to determine the optimal combination of multi-parameter ranges for improving the calibration of hydrological models. At first, single-parameter probability distributions were analysed based on 100 samples obtained from independent Genetic Algorithms (GA) calibration performed on a Xinganjiang model with a corresponding initial parameter range and, the distribution type (i.e. normal, exponential and uniform distributions) was specified for each parameter of the model. Then, the optimal range for each parameter was determined by comparing $E_{NS}$ values calculated separately with the initial range, the minimum and maximum ranges of a given cumulative frequency of 50% (i.e. MINR and MAXR) and the extended range. Next, parameter correlation and sensibility were evaluated by quantifying two indexes $R_{C\,Y,X}$ and $S_E$ which can be used to coordinate with the negatively correlated parameters to specify the optimal combination of ranges of all parameters for calibrating models. It is shown from the investigation that the probability distribution of calibrated values of any particular parameter in a Xinanjiang model is closely approximated by a normal or exponential distribution. The multi-parameter optimal range selection method is superior to the single-parameter one for calibrating hydrological models with multiple parameters. The combination of optimal ranges of all

parameters is not the optimum inasmuch as some parameters like Im have negative effects on other parameters. The application of the proposed methodology gives a rise to an increase of 0.01 in minimum $E_{NS}$ compared with that of the pure GA method. The rising of minimum $E_{NS}$ with little change of the maximum may shrink the range of the possible solutions, which can effectively reduce uncertainty of the model performance."

**14.** Following "Referee #2 Specific comment #2" comments, we have replaced "method" with "tool" in the first sentence of "Introduction". The following were added and/or modified:

Page 1, line 1:

"Hydrological process modelling is an important tool for research on water resources management …"

**15.** Following "Referee #2 Specific comment #4" comments, we have modify it as it is suggested. The following were added and/or modified:

Page 4, line 30:

"A Genetic Algorithm (GA) was selected …"

**16.** Following "Referee #2 Specific comment #5" comments, we have modify the sentence to make it easy to read. The following were added and/or modified:

Page 9, line 6:

"It is obvious from Table 4 that …"

**17.** Following "Referee #2 Specific comment #6" comments, we have removed the colon as it is suggested. The following were added and/or modified:

Page 10, line 7:

"… findings are presented as follows:">>"… findings are presented as follows."

**18.** Following "Referee #2 Specific comment #8" comments, we have the label "schema" in Figs. 6a, 6b, 6c, 7a, 7b, 7c, 8a, 8b, 8c and 8d. The following were added and/or modified:

[Figure]

Fig. 8. Results of range selection of parameter CI

Probability distribution of parameter values for schema initial range (a), CI-MINR (b) and CI-MAXR (c);

(d) Box-plot chart of $E_{NS}$ for three schemas

[Figure]

Fig. 9. Results of range selection of parameter KI

Probability distribution of parameter values for schema initial range (a), KI-MINR (b) and KI-MAXR (c);

(d) Box-plot chart of $E_{NS}$ for three schemas

[Figure]

Fig. 10. Results of range selection of parameter B

Probability distribution for schema initial range (a), B–Extension (b), B–MINR (c) and B–Extension–MINR (d);

(e) Box–plot chart of $E_{NS}$ for four schemas

**19.** Following "Referee #2 Specific comment #9" comments, we have modify the note on Table 1 to make it easy to read. The following were added and/or modified:

Page 22, note on Table 1:

[revised manuscript text omitted]

---

## Author Response (AR2)

**Part 1. Response to Referee's Comments**

**Responses to Referee #1**

**Main comment #1**

Even if the language was generally improved in this revised version some sections still require further work. More specifically, the Introduction section needs most work. This is not a trivial issue as it affects the readability of the manuscript and can discourage potential readers. I would recommend the authors to have a native English speaker proof-check the manuscript.

**Responses to main comment #1:**

A native English speaker has checked the language in the revised paper. For more details, please see Part 2. "Relevant Changes Following Referee's Comments".

**Main comment #2**

In the revision the authors clarified the magnitude of the efficiency improvement produced by the proposed method. In their response to my previous comment the authors recognize that the modes performance increase might be due to use of the Xinanjiang model for the case study. As I see it a case study should be selected to demonstrate the benefits of the proposed methodology. Following this reasoning I wonder the rationale behind selecting the specific catchment and hydro-meteorological model for this study. Please clarify.

**Responses to main comment #2:**

Since the Xinanjiang model, as a concept model, is applicable for humid and semi-humid regions like South China, it was adopted to simulate the rainfall-runoff process in the Chaotianhe River basin with mean annual precipitation of 1700 mm. Moreover, there are a few parameters of negative correlations with other parameters (e.g. Im and EX, SM and B) in the Xinanjiang model, which affects the search for the optimal parameter range. Another reason for choosing the Xinanjiang model is to prove that the proposed methodology can deal with the contradiction between the ranges selection of multiple parameters. Karst areas are of special hydro-geology features and hence the values of some parameters of hydrology models quite different with those for normal areas. For instance, parameter B in the Xinanjiang model, which represents the non-uniformity of storage condition in space, is generally ranging from 0.2 to 0.3 for normal areas of 100-1000 km$^2$, while approximate to 0.4 for karst areas of a similar size according to the previous studies. It is found from the investigation that the optimal range of parameter B is 0.379-0.488 for the Chaotianhe River basin (a karst region) of 476 km$^2$. We chose a karst basin for the case study to demonstrate the correctness of the proposed methodology.

**Main comment #3**

Page 4, line 8. The authors report that areal precipitation was calculated based on the Thiessen polygon method. This method is known to introduce significant errors in catchments with complex topography. Since the study area appears to be quite mountainous I wonder about the rationale behind using the Thiesse polygon method. Please clarify.

**Responses to main comment #3:**

We agree with the opinion of the referee. The Thiessen polygon method is not applicable for the areas with complex topography since it is difficult to handle the effect of topography on rainfall. However, about 80% of the study basin is of the elevation ranging from 150 to 800 m above the sea level and the highest is about 1687 m, which implies that the terrain change is not very large for the study basin compared with those for other high-altitude mountain basins. Moreover, four precipitation stations are located at different altitudes and hence the data collected can roughly reflect the effect of elevation on rainfall. In addition, considering that there is no precipitation station in the adjacent basin, interpolation methods maybe introduce larger errors based on the data collected from four stations. These are the reasons why we adopted the Thiesson polygon method to calculate the areal precipitation. Anyway, we appreciate the suggestion of the referee and will do more work to improve the calculation precision of areal precipitation in future research.

[Figure]

Figure A The topography of the Chaotianhe River basin

**Specific comment #1**

Page 4, line 10: The most common and accepted term is "hydro-meteorological ".

**Responses to specific comment #1:**

"metro-hydrological" has been replaced with "hydro-meteorological".

**Responses to Referee #2**

**Main comment #1**

Abstract is not well structured and not clear. Hereafter some comments. The sentence at line 16 is not easy to read. I suggest to limit it as "…initial parameter range and distribution type (i.e. normal, exponential and uniform distributions)." In the following sentences Authors refer to specific indexes, such as ENS, MINR, MAXR that are not know by the readers. They cannot guess their meaning. This happens again at line 25 with Im. Please, revise the overall abstract in order to summarize the concepts of your work, the elements of novelty but without referring to specific parameters or indexes that are defined only in the main text.

**Responses to main comment #1:**

We have removed some elements referring to specific parameter (Im) and indexes (MINR and MAXR). And we have added the complete description of $E_{NS}$. We have rewritten the abstract as follows:

The parameters are usually calibrated to achieve good performance of hydrological models, owing to the highly non-linear problem of hydrology process modelling. However, parameter calibration efficiency has a direct relation with parameter range. Furthermore, parameter range selection is affected by probability distribution of parameter values, parameter sensitivity and correlation. A newly proposed method is employed to determine the optimal combination of multi-parameter ranges for improving the calibration of hydrological models. At first, the probability distribution (i.e. normal, exponential and uniform distributions) was specified for each parameter of the model based on Genetic Algorithms (GA) calibration. Then, several ranges were selected for each parameter according to the corresponding probability distribution and subsequently the optimal range was determined by comparing the model results calibrated with the different selected ranges. Next, parameter correlation and sensibility were evaluated by quantifying two indexes $R_{C\,Y,X}$ and $S_E$ which can be used to coordinate with the negatively correlated parameters to specify the optimal combination of ranges of all parameters for calibrating models. It is shown from the investigation that the probability distribution of calibrated values of any particular parameter in a Xinanjiang model approaches a normal or exponential distribution. The multi-parameter optimal range selection method is superior to the single-parameter one for calibrating hydrological models with multiple parameters. The combination of optimal ranges of all parameters is not the optimum inasmuch as some parameters have negative effects on other parameters. The application of the proposed methodology gives rise to an increase of 0.01 in minimum Nash-Sutcliffe efficiency ($E_{NS}$) compared with that of the pure GA method. The rising of minimum $E_{NS}$ with little change of the maximum may shrink the range of the possible solutions, which can effectively reduce uncertainty of the model performance.

**Specific comment #1**

P3,L0: special? I think "specific" here is more appropriate.

**Responses to specific comment #1:**

We have replaced "special" with "specific".

**Specific comment #2**

P10, L24: I am still not convinced that a variation in term of model efficiency of nearly 0.01 could be defined as "much greater". I do agree with the importance of reducing the NSE variability, but I suggest to reduce the emphasis on this point.

**Responses to specific comment #2:**

We have replaced "much greater" with "more robust".

**Specific comment #3**

P10, L33: "maximum and minimum ENS values…" and " using the proposed" (please check for this in many cases along the manuscript); please remove "respectively"

**Responses to specific comment #3:**

We have removed "respectively" in: 1) P10, L33, 2) P6, L9, 3) P9, L20, 4) P9, L27.

**Specific comment #4**

Figure 12: I appreciate the new figure but I wonder why there is such a brutal change on the min values after 100 runs. It's a bit strange that GA and the proposed method produce essentially the same values till 100 run and then they suddenly diverge. Do the Authors have any explanation for that? Can it be related to the sampling number of parameter values? What about the computational time required by the two methodologies? Are there significant differences?

**Responses to specific comment #4:**

In the paper, it is reported that GA and the proposed method produce the same values till 100 runs and then they suddenly diverge. It is because that the proposed PRS method initially needs 100 runs of GA calibration to obtain parameter value samples for selecting the optimal ranges. In addition, there is no significant difference found in computation time between two methodologies. As it is shown in Figure B, the total computational times for 100 runs vary little with study cases. We have added the related response in the last paragraph of Section 4.3.

[Figure]

Figure B. The total computational time for 100 runs in different cases

**Specific comment #5**

Conclusion: L10, please remove "for karst area", it's specify at the previous line.

**Responses to specific comment #5:**

We have removed "for karst area".

**Part 2. Relevant Changes Following Referee's Comments**

1. Following "Referee #1 Main comment #1" and "Referee #2 general comment", we have checked the text of the manuscript. The following were added and/or modified:

   1) Page 2, line 5-15:
   "… distributed models …" >> "… physically-based models"

   2) Page 2, line 6:
   "… improved in recent years and their structures become more and more mature." >> "… improved in recent years with their structures becoming more mature."

   3) Page 2, line 7:
   "…the distributed models have definite physical mechanism …" >> "…the physically-based model has definite physical mechanism…"

   4) Page 2, line 11:
   "…models have better performance of modelling …" >> "…models have better performance in modelling …"

   5) Page 2, line 20:
   "… mathematical calculations, having a wide application in …" >> "…mathematical calculations and thus more widely applied in …"

   6) Page 2, line 23:
   "…useful methods, but has its limitation of initial value ranges of parameters …" >> "…useful methods, but at the same time has a negative side of being bounded by initial value ranges of parameters …"

   7) Page 2, line 25-26:
   "…the Genetic Algorithm (GA) is of random search strategy that avoids problem of local search, being a global optimization algorithm in a real sense …" >> "…the Genetic Algorithm (GA) which is designed with random search strategy can avoid the problem of local search, thus is a global optimization algorithm in its essence …"

   8) Page 2, line 31:
   "…Though the auto-calibration method …" >> "…Although the auto-calibration method …"

   9) Page 3, line 1:
   "…parameter variables obey some special probability distributions …" >> "…parameter variables follow some specific probability distributions …"

   10) Page 3, line 5:
   "…probability distribution can be provide a clue to realize …" >> "…probability distribution can provide a clue to realize …"

   11) Page 3, line 6:
   "…probability distributions seldom were investigated …" >> "…probability distributions were seldom investigated …"

   12) Page 3, line 8:
   "…sensitive characteristics and correlation behavior …" >> "…sensitive characteristics and correlation patterns…"

   13) Page 3, line 9:

"…should be calibrated, but the insensitive parameter …" >> "…should be calibrated, while the insensitive parameter …"

14) Page 3, line 15:
"…might make insensitive parameters become sensitive ones …" >> "…might render insensitive parameters into sensitive ones …"

15) Page 3, line 16:
"… be taken into considered when the calibration of multi-parameters models…" >> "…be taken into consideration when the calibration of multi-parameter models…"

16) Page 3, line 20:
"…few literature covers information on how to select …" >> "… few literature has been documented on how to select …"

17) Page 3, line 25:
"…based on a lot of independent calibrations …" >> "…based on many independent calibrations …"

18) Page 4, line 6:
"The daily data concerning precipitation …" >> "The data concerning daily precipitation …"

19) Page 4, line 11-12:
"…streamflow is about 719 $m^3$/s, the minimum 0.53 $m^3$/s and the average is 13.31 $m^3$/s …" >> "…streamflow was about 719 $m^3$/s, the minimum 0.53 $m^3$/s and the average was 13.31 $m^3$/s …"

20) Page 4, line 13:
"…precipitation varied with years in the studied catchment …" >> "…precipitation varies with years in the studied catchment …"

21) Page 4, line 26-27:
"…each other but obey a certain probability distribution …" >> "…each other but follow a specific probability distribution …"

22) Page 4, line 28-29:
"…In order to analyse the probability distribution of parameter values, a stochastic-based auto-calibration is used to calibrate the model, and samples of calibrated parameters values are obtained …" >> "…The stochastic-based auto-calibration is used to calibrate the model, and samples of calibrated parameters values are obtained in order to analyse the probability distribution of parameter values …"

23) Page 4, line 29-30:
"…As far as the sample size is concerned, 100 samples are enough to estimate the probability distribution …" >> "…The sample size of 100 is adequate for to estimating the probability distribution …"

24) Page 4, line 31:
"…keep stable when sampling size is …" >> "…keep stable when sample size is …"

25) Page 5, line 23:
"… (K-S test) tries to examine whether a data set fit a reference…" >> "…(K-S test) is geared to examining whether a data set fits a reference …"

26) Page 6, line 3-4:
"…In consideration of the three probability distribution types …" >> "… Given the three

probability distribution types …"

27) Page 6, line 5:

"…due to little influence of ranges on …" >> "…due to the weak influence of ranges on …"

28) Page 6, line 15:

"… maximum $E_{NS}$ keeps constant despite …" >> "…maximum $E_{NS}$ remains constant despite …"

29) Page 6, line 25:

"… The correlations among parameters, therefore, should be taken into account, if the ranges of the related parameters required adjusting…" >> "…If the ranges of the related parameters required adjusting, the correlations among parameters, therefore, should be taken into account. …"

30) Page 6, line 28-29:

"…model calibration, although the optimal ranges of …" >> "…model calibration, even though the optimal ranges of …"

31) Page 6, line 32:

"…is minus, it means a negative influence…" >> "…is minus, it exerts a negative influence…"

32) Page 7, line 7:

"…other parameter kept for calibration generally to mitigate …" >> "…other parameter is kept for calibration generally to mitigate …"

33) Page 7, line 7-8:

"…It is due to that sensitive parameters…" >> "…It is due to the fact that sensitive parameters…"

34) Page 7, line 11:

"…given priority to use the optimal range …" >> "…given priority to the optimal range…"

35) Page 7, line 16:

"…Considering there are more than…" >> "…Given the fact that there are more than …"

36) Page 7, line 20:

"…a set of initial ranges of parameters is given for…" >> "…a set of initial ranges of parameters are given for…"

37) Page 8, line 14-15:

"…the box-plots, it is shown that the probability distributions …" >> "…the box-plots, the probability distributions …"

38) Page 8, line 23-24:

"…It implies that reducing the initial ranges can improve the calibration for parameters whose values obey normal distributions …" >> "…It indicates that reducing the initial ranges can improve the calibration for parameters whose values observe normal distributions…"

39) Page 8, line 28:

"…reducing the range was used to find the optimal range …" >> "…the range was reduced to find the optimal range…"

40) Page 9, line 8:

"…picked out by using the cumulative frequency curve…" >> "…picked out according to the cumulative frequency curve…"

41) Page 9, line 26:

"…are found when the initial range is substituted with the three ranges respectively …" >> "…are found when they replace the initial range …"

42) Page 10, line 1:

"…small values, which implies that using the optimal ranges…" >> "…small values, indicating that using the optimal ranges …"

43) Page 10, line 2:

"…is not conductive to calibrate these …" >> "…is not conductive to calibrating these…"

2. Following "Referee #1 Specific comment #1", we replaced "metro-hydrological" with "hydro-meteorological". The following were added and/or modified:

Page 4, line 10:

"Some metro-hydrological statistical data …" >> "Some hydro-meteorological statistical data …"

3. Following "Referee #2 Main comment #1", we modified the abstract as suggestion. The following were added and/or modified:

a. Page 1, line 14-19:

"At first, single-parameter probability distributions were analysed based on 100 samples obtained from independent Genetic Algorithms (GA) calibration performed on a Xinganjiang model with a corresponding initial parameter range and, the distribution type (i.e. normal, exponential and uniform distributions) was specified for each parameter of the model. Then, the optimal range for each parameter was determined by comparing $E_{NS}$ values calculated separately with the initial range, the minimum and maximum ranges of a given cumulative frequency of 50% (i.e. MINR and MAXR) and the extended range." >> "At first, the probability distribution (i.e. normal, exponential and uniform distributions) was specified for each parameter of the model based on Genetic Algorithms (GA) calibration. Then, several ranges were selected for each parameter according to the corresponding probability distribution and subsequently the optimal range was determined by comparing the model results calibrated with the different selected ranges."

b. Page 1, line 22-23:

"… is closely approximated by a normal or exponential distribution." >> "…approaches a normal or exponential distribution."

c. Page 1, line 25:

"… parameters like Im have negative effects on …" >> "… parameters have negative effects on …"

d. Page 1, line 26:

"… give a rise to an increase of 0.01 in minimum $E_{NS}$ …" >> "… give rise to an increase of 0.01 in minimum Nash-Sutcliffe efficiency ($E_{NS}$) …"

4. Following "Referee #2 Specific comment #1", we replaced "special" with "specific". The following were added and/or modified:

Page 3, line 1:

"… obey some special probability distributions …" >> "… follow some specific probability distributions …"

5. Following "Referee #2 Specific comment #2", we replaced "much greater" with "more robust". The following were added and/or modified:

Page 10, line 24:

"…the $E_{NS}$ values are much greater than those of …" >> "… the $E_{NS}$ values are more robust than those of …"

6. Following "Referee #2 Specific comment #3", we removed some "respectively" as suggestion. The following were added and/or modified:

   a. Page 10, line 32-33:

"… using a GA method and a proposed PRS method, respectively." >> "… using a GA method and a proposed PRS method."

   b. Page 6, line 9:

"… under a given cumulative frequency, respectively." >> "… under a given cumulative frequency"

   c. Page 9, line 20.

"After the MINR selection was performed on the initial range and the extended range respectively, the two ranges …" >> "After the MINR selection was performed on the initial range and the extended range, the two ranges …"

   d. Page 9, line 27:

"…are found when the initial range is substituted with the three ranges respectively…" >> "…are found when they replace the initial range …"

7. Following "Referee #2 Specific comment #4", we added some explanation for the brutal change on the min values after 100 runs in Fig. 12. The following were added and/or modified:

   a. Page 10, line 33-34

"It is indicated from Figure 12 that no mater it is maximum or minimum $E_{NS}$, the value calculated by using a proposed method is almost the same as that by using a GA method when the number of runs does not exceed 100." >> "It is indicated from Fig. 12 that no mater it is maximum or minimum $E_{NS}$, the PRS-based value is essentially the same as the GA-based one when the number of runs does not exceed 100. It is because that the PRS method initially need 100 runs of GA calibration to obtain parameter value samples for selecting the optimal ranges."

   b. Page 11, line 3-4:

"… by using a GA method. The application of a proposed …" >> "… by using a GA method. Additionally, it is found from the investigation that there is no significant difference in computational time between the two methodologies. The application of a proposed …"

8. Following "Referee #2 Specific comment #5", we removed "for karst area" as suggestion. The following were added and/or modified:

Page 11, line 10:

"In the Xinanjiang model for karst areas, …" >> "In the Xinanjiang model, …"

**Improvement of hydrological model calibration by selecting multiple parameter ranges**

Qiaofeng Wu[1], Shuguang Liu[1,2], Yi Cai[1,2], Xinjian Li[3] and Yangming Jiang[4]

[1]Department of Hydraulic Engineering, College of Civil Engineering, Tongji University, Shanghai, 200092, China
5  [2]The Yangtze River Water Environment Key Laboratory of the Ministry of Education, Tongji University, Shanghai, 200092, China
[3]Irrigation Experiment Center of Guangxi Zhuang Autonomous Region, Guilin, 541105, China
[4]Hydrology & Water Resources Bureau of Guangxi Zhuang Autonomous Region, Guilin, 541001, China

*Correspondence to*: Yi Cai (caiyi@tongji.edu.cn)

10  **Abstract.** The parameters are usually calibrated to achieve good performance of hydrological models, owing to the highly non-linear problem of hydrology process modelling. However, parameter calibration efficiency has a direct relation with parameter range. Furthermore, parameter range selection is affected by probability distribution of parameter values, parameter sensitivity and correlation. A newly proposed method is employed to determine the optimal combination of multi-parameter ranges for improving the calibration of hydrological models. At first,  the probability distribution
15   was specified for each parameter of the model based on Genetic Algorithms (GA) calibration. Then, several ranges were selected for each parameter according to the corresponding probability distribution and subsequently the optimal range was determined by comparing the model results calibrated with the different selected ranges.
20   Next, parameter correlation and sensibility were evaluated by quantifying two indexes $R_{C\,Y,X}$ and $S_E$ which can be used to coordinate with the negatively correlated parameters to specify the optimal combination of ranges of all parameters for calibrating models. It is shown from the investigation that the probability distribution of calibrated values of any particular parameter in a Xinanjiang model
25   approaches a normal or exponential distribution. The multi-parameter optimal range selection method is superior to the single-parameter one for calibrating hydrological models with multiple parameters. The combination of optimal ranges of all parameters is not the optimum inasmuch as some parameters  have negative effects on other parameters. The application of the proposed methodology gives  rise to an increase of 0.01 in minimum Nash-Sutcliffe efficiency ($E_{NS}$) compared with that of the pure GA method. The rising of minimum $E_{NS}$ with little change of the maximum
30  may shrink the range of the possible solutions, which can effectively reduce uncertainty of the model performance.

**Key words**: hydrological model, calibration, parameter ranges, probability distribution

**1. Introduction**

Hydrological process modelling is an important tool for research on water resources management, flood control and disaster mitigation, water conservancy project planning and design, hydrological response to climate change and so on (Zanon et al., 2010;Papathanasiou et al., 2015). The initial hydrological model was a black-box model in 1932 (Sherman, 1932) and conceptual & physically-based models are subsequently put forward in 1960s (Freeze and Harlan, 1969). The three kinds of hydrological models have been significantly improved in recent years  with their structures becoming more  mature. Theoretically, the physically-based models have definite physical mechanism of the water cycle and all parameters can be measured in-situ (Abbott et al., 1986;Huang et al., 2014). Conceptual models express hydrological processes in form of some abstract models which come from some physical phenomenon and experience. For example, the interflow and the base flow are simplified as the flow from linear reservoirs (Caviedes-Voullième et al., 2012;Lü et al., 2013). As a result, some parameters of conceptual models need calibrating. In general, conceptual models have better performance in  modelling the streamflow at the catchment outlet than physically-based distributed models do, especially for catchments lacking sufficient data (Bao et al., 2010;Cullmann et al., 2011). Thus, many conceptual models such as HBV model, TOPMODEL, Tank model and Xinanjiang model are of strong vitality (Abebe et al., 2010;Vincendon et al., 2010;Hao et al., 2015; Xie et al., 2015). Additionally, the performance of physically-based distributed models can be improved after calibration of some parameters (Chen et al. 2016). Therefore, all of the hydrological models should be calibrated before engineering applications.

There are two kinds of calibration methods for hydrological models, the trial-error method and auto-calibration method. The trial-error method depends on plenty of trials for reducing the error of the objective. However, it is difficult to obtain an exact optimal solution due to limited enumeration (Boyle et al., 2000). The auto-calibration method is based on stochastic or mathematical calculations and thus more widely applied in the non-linear parameter optimization. Compared with the trial-error method, it is more efficient and effective, avoiding the interference of anthropogenic factors (Madsen, 2000;Getirana, 2010). The initial automatic optimization methods, such as the Rosenbrock Method (Rosenbrock, 1960) and the Simplex Method (Nelder and Mead, 1965), are classical and useful methods, but at the same time has a negative side of being bounded by initial value ranges of parameters. Therefore, it can only be regarded as local optimization algorithms (Gupta and Sorooshian, 1985). Different from classical methods above, the Genetic Algorithm (GA) which is designed with random search strategy  can avoid the problem of local search, thus is a global optimization algorithm in its essence (
[revised manuscript text omitted]
 100. It is because that the PRS method initially need 100 runs of GA calibration to obtain parameter value samples for selecting the optimal ranges. If a proposed method is used for calibration instead of a GA method, there are approximately an increase of 0.001 in maximum $E_{NS}$ and an increase of 0.01 in minimum $E_{NS}$ when the number of runs is greater than 100. Thus, for any particular run number, the value of $E_{NS}$ calculated by using a PRS method is not less than that by using a GA method. Additionally, it is found from the investigation that there is no significant difference in computational time between the two methodologies. The application of a proposed method, therefore, contributes to a relatively efficient calibration.

**5. Conclusions**

Considering that there is a relation between the selection of multi-parameter ranges and the calibration effect of a hydrological model, an approach to determine an optimal combination of ranges for the multi-parameter calibration was put forward by analysing parameter probability distribution, parameter sensitivity and correlation between parameters. The newly proposed method was applied for the calibration of a Xinanjiang model for karst areas, and some findings are presented as follows.

[revised manuscript text omitted]